# A Review of the Authentication Techniques for Internet of Things Devices in Smart Cities: Opportunities, Challenges, and Future Directions

**DOI:** 10.3390/s25061649

**Published:** 2025-03-07

**Authors:** Ashwag Alotaibi, Huda Aldawghan, Ahmed Aljughaiman

**Affiliations:** Department of Computer Networks and Communications, College of Computer Sciences and Information Technology, King Faisal University, Al-Ahsa 31982, Saudi Arabia

**Keywords:** Internet of Things, authentication, smart cities, blockchain, edge computing

## Abstract

Smart cities have witnessed a transformation in urban living through the Internet of Things (IoT), which has improved connectedness, efficiency, and sustainability. However, the adoption of IoT devices presents significant security vulnerabilities, particularly in authentication. The specific limitations of IoT contexts, such as constrained computational resources, are frequently not adequately addressed by traditional authentication techniques. The existing methods of authentication used for IoT devices in smart cities are critically examined in this review study. We evaluate the advantages and disadvantages of each mechanism, emphasizing real-world applicability. Additionally, we examine cutting-edge developments that offer improved security and scalability, such as blockchain technology, biometric authentication, and machine learning-based solutions. This study aims to identify gaps and propose future research directions to develop robust authentication frameworks that protect user privacy and data integrity.

## 1. Introduction

A new era of urban living marked by increased sustainability, better resource management, and improved connection has been brought about by the incorporation of the IoT into smart cities [1]. IoT devices are crucial in changing several industries, such as public safety, energy management, and transportation [2]. However, there are significant security issues related to IoT device authentication. Authentication is essential to ensure authorized access [3]. Because of the limitations of low-power devices and the ongoing risk of cyberattacks, traditional authentication techniques, including username–password combinations, frequently prove insufficient in the context of the IoT [4]. Man-in-the-middle (MitM) attacks, replay attacks, and credential theft are just a few of the vulnerabilities that highlight the need for stronger authentication solutions. The objective of this systematic literature study is to assess the efficacy and usability of the authentication methods currently used in IoT devices in smart cities. This study looks at recent developments including blockchain technology, biometric authentication, and machine learning-based techniques to find research gaps and suggest future paths for improving IoT security. The ultimate goal of this effort is to aid in the creation of robust authentication frameworks that can safeguard user confidentiality and data integrity while guaranteeing the safe functioning of IoT networks in intelligent urban environments.

### 1.1. Background

The IoT is an advanced technology that connects multiple digital devices equipped with multiple sensing, actuation, and processing capabilities to the Internet, enabling several new services. As an example, smart lighting systems can improve public safety and energy efficiency by automatically adjusting brightness based on real-time car and human presence. Furthermore, smart parking systems make real-time parking availability monitoring possible, which also helps ease traffic by directing vehicles to available spaces. Additionally, IoT devices are used in smart traffic management to maximize traffic flow, reduce delays, and boost public transportation system effectiveness. IoT has significantly improved urban quality of life and services [1]. Below are some key points to illustrate the importance of the IoT in improving and developing smart cities.

Quality of Life

IoT devices can enable residents to better control the security systems and energy consumption of their smart homes [2].

Enhanced Urban Services

Services such as public safety, energy management, and transportation have been improved through IoT applications, such as smart traffic systems that have been used to reduce traffic congestion and intelligent waste management solutions that enhance collection routes by using data in real time [2,5].

Sustainability and Environmental Monitoring

IoT applications enable the monitoring of air quality and noise levels, which greatly helps in responding to environmental conditions. This develops the ability of smart cities to respond effectively to challenges related to environmental conditions [2].

Smart devices have become essential to the IoT. They have complete control over real-time physical object monitoring and can offer citizens intelligent information. Smart devices, including actuators and sensors, gather data at the network edge to facilitate decision-making [3]. In smart city architecture, the application of these smart gadgets spreads to various physical and cyber systems, such as smart homes, smart grids, smart transportation, smart healthcare, smart agriculture, smart buildings, public safety, smart parking, smart traffic systems, etc. The security of these devices is crucial due to varying architectural challenges. There are concerns about data privacy and cybersecurity risks, especially concerning authentication, as the entire infrastructure is at risk in the event of unauthenticated or malicious assets. These concerns must be addressed during development and deployment [2].

### 1.2. Motivation

Many objects are now connected to the Internet. These objects can autonomously sense, act, and convey information. This is known as IoT technology. The interconnected nature of devices within the IoT renders them vulnerable to cyber threats, including unauthorized access. There are many cybersecurity threats related to IoT, specifically related to authentication. Protocols like MQTT may perform inadequate verification, allowing attackers to intercept and tamper with messages. Also, one of these threats is MitM attacks [3]. The use of such attacks occurs when an attacker intercepts communications or messages between clients or servers, and this happens without the knowledge of one of the parties and is due to the lack of proper authentication. Also, there exist replay attack threats, through which the attacker obtains authentication messages for unauthorized access because IoT environments lack strong mechanisms for devices to distinguish between new and replayed messages. In addition, improper processing of messages in protocols such as CoAP may lead to buffer overflow, which exposes network devices to risk and may also lead to Access Control Failures [6]. Another potential threat is the physical access to IoT devices, such as sensors which could be tampered with or changed, allowing important data to be accessed and the whole network to be manipulated without authorization, which puts everyone at risk. Also, through fake node attacks, the attacker can introduce a malicious device into the network; this device sends inaccurate and fake data that disrupt normal operations, drain the resources of legitimate devices, and overload them, which leads to a DoS [4]. This research emphasizes the critical role of authentication in restricting unauthorized access to interconnected devices.

### 1.3. Problem Statement

Smart city deployments of IoT devices can transform urban living through increased automation and better resource management. However, this quick integration brings up serious security issues, especially concerning IoT device authentication. Authentication is essential to ensure that only authorized devices may access networks and services. Various methods exist for authentication, each with limitations. One of the most common methods is to provide a username and password for users or devices. Still, this method is vulnerable to brute force attacks or phishing attacks and is ineffective if the goal is to ensure strong security in IoT environments because users often use weak passwords and do not change them regularly. Lightweight authentication protocols are designed for resource-constrained environments. While aimed at efficiency, these protocols often sacrifice security, leaving IoT devices vulnerable. Blockchain-based authentication offers a decentralized solution but can introduce latency and complexity [7].

It is also possible to use the biometric authentication method. This includes fingerprint scanners and facial recognition. While secure, biometric data can be captured and misused [5]. If the biometric data are hacked, they cannot be changed. Therefore, when storing biometric data, there are strict security measures to prevent these violations [4]. PKI uses public and private keys for secure authentication but poses challenges for low-power IoT devices due to high computational costs. Also, in large-scale distribution operations, maintaining and distributing these keys securely is very tiring. It is possible to use challenge–response protocols, in which the verifier sends a challenge to which the client must respond correctly to prove its identity. In this method, if the challenges are not unique or time-sensitive, the attacker may be able to replay these valid responses and gain unauthorized access [8].

In this method, there is also a high communication overhead [8]. It is possible to use cognitive and behavioral methods that are based on certain personal habits, such as typing cadence or eye movement. In this approach, there may be additional sensors and complex algorithms that may not apply to all IoT devices. There may also be inconsistencies in reliability because user behavior may change over time, whether due to stress, fatigue, or other factors [9]. The urgent need for creative and practical authentication techniques designed specifically for IoT devices in smart cities is highlighted by this issue statement. To guarantee the security and resilience of smart urban settings, these issues must be resolved.

### 1.4. Scope

This research aims to investigate and provide reliable authentication methods for the IoT devices that are placed in smart city environments. As smart cities depend more and more on networked devices, it is critical to ensure data integrity and safe access. In this research, we focus on the basic aspects, which are the authentication methods used in IoT devices, the types of IoT devices, and the potential resource constraints in a smart city environment. A major challenge IoT devices face in smart cities is data volume, complicating scalability. When it comes to storing and processing this amount of data, traditional storage solutions are not enough; scalable cloud storage solutions are needed that process data in real time. Real-time data processing is essential, but increased device and data volume challenges scalability [10]. Resource constraints may affect the effectiveness of IoT devices, especially in smart cities, such as energy consumption, as some IoT devices operate on batteries, which have limited capacity. Therefore, it is necessary to provide innovative energy-saving technologies such as WPT technologies to extend the life of devices [11,12]. Focusing on device security is crucial, as limited processing capabilities can hinder strong security implementation, leading to vulnerabilities [10,11]. Blockchain technology and edge computing enhance IoT effectiveness, improving operational efficiency and security.

Blockchain technology enhances trust and security in IoT applications through decentralized networks that verify transactions. Following data entry onto the blockchain, data cannot be modified or even deleted without approval, making it difficult for malicious parties to manipulate records. To enhance accountability, blockchain works on transparent transactions, meaning that all transactions are visible to participants and all actions can be tracked to ensure their legitimacy. Each transaction is encrypted through cryptographic hash to keep data protected from unauthorized access. The technology also reduces points of failure, which enhances the system’s overall security [13].

Edge computing minimizes latency for real-time applications like traffic control by processing data near its source. IoT gadgets in smart cities also produce a lot of data, so improving bandwidth by reducing its consumption helps relieve network congestion, which is what edge computing technology does [13]. Privacy can be enhanced by retaining sensitive information locally instead of sending it to the cloud using Edge computing [10]. Therefore, this project provides useful insights that enhance IoT device security in smart cities, ensuring that they operate safely and effectively.

### 1.5. Novelty of the Project

This literature review aims to analyze the current authentication mechanisms and their latest developments that have been utilized in IoT devices in smart cities. The rising quantity of these gadgets in the present era has made ensuring their security of utmost importance. This review helps to gain a deeper understanding of the challenges facing authentication mechanisms in urban environments as it integrates ideas about cybersecurity, the IoT, and urban environments to provide a comprehensive perspective that helps researchers in this field. The project aims to provide insights for researchers interested in enhancing security in smart cities by highlighting the current challenges facing IoT authentication. The project not only summarizes the existing literature but also helps identify gaps in current research and suggests future research directions for progress. The goal of this research is to help the development of the IoT system to make it more efficient in smart cities.

### 1.6. Research Contributions

Review the current state of authentication mechanisms used in IoT devices, focusing on smart cities.Draw attention to the main issues with IoT authentication, such as compatibility and scalability.Evaluate the effectiveness, usability, and security of various authentication methods such as password-based, biometric, and other methods in smart cities.Assess the effect of authentication strategies on smart city security and privacy.Provide information regarding IoT authentication practices to users and emphasize the need for user awareness and education.

## 2. Attacks on IoT Authentication

IoT systems are vulnerable to several attacks that target IoT authentication techniques. Figure 1 shows the common types of attacks, including network attacks, device attacks, authentication attacks, protocol attacks, and data attacks.

### 2.1. Network Attacks

Along with IoT advantages, there are also significant cybersecurity risks associated with this transformation. There are several types of network attacks to be aware of when it comes to IoT authentication in smart cities, the most common of which are MitM, DoS, and packet sniffing [14].

#### MitM

IoT authentication systems are seriously threatened by MitM attacks, especially when it comes to smart cities. MitM attacks take place when a hacker secretly intercepts and maybe modifies two-party communications [15].

There are several types of MitM attacks related to the IoT in smart cities, the most famous of which are Wi-Fi eavesdropping, DNS spoofing session hijacking, SSL stripping, ARP spoofing, and cloud polling attacks. Figure 2 illustrates the types of MitM attacks in IoT authentication. Attackers may intercept any data being transferred, including login passwords and private information, when users or devices connect to these networks. DNS spoofing is an attack technique whereby the attacker modifies DNS records to reroute consumers to malicious websites that look authentic. Session hijacking is the practice of taking session cookies from an IoT application after a user has logged in. Adversaries can convert secure HTTPS connections back to unsecured HTTP connections using SSL stripping. In ARP Spoofing, attackers can link their MAC address with the IP address of a legitimate device by using a local network to transmit bogus ARP packets. In cloud polling attacks, to obtain new information and updates, many IoT devices in smart cities communicate with cloud services. Through this communication, the attacker intercepts it, especially if the devices do not authenticate the cloud service, which may lead to data breaches [16,17,18].

First and foremost, a major reason why IoT devices are vulnerable to eavesdropping is that they use outdated or insecure communication protocols with weak encryption. Moreover, different manufacturers sometimes offer devices with varying security. Additionally, a huge attack surface is created, making it difficult to effectively monitor and secure all communications. Furthermore, many IoT devices do not receive regular upgrades or security patches. Creating secure authentication against MitM attacks is a challenging task due to the complex environment created by all these elements [10,19].

There are several ways to combat MitM attacks on IoT authentication in smart cities. First and foremost, using TLS, a type of strong encryption that ensures secure transmission of data between IoT devices, makes these data protected from eavesdropping. Also, using PKI, a strong authentication technology, can help prevent unwanted access. In addition, regular updates are necessary to fix known vulnerabilities and guarantee the security of IoT devices and software. Moreover, using anomaly detection systems with continuous network monitoring helps identify and detect anomalous traffic patterns indicating that there are MitM attacks to help and allow for a quick response [10,20].

### 2.2. Authentication Attacks

Malicious attempts to compromise authentication systems, which are essential for protecting private data and limiting access to just those who are permitted to certain resources, are known as authentication attacks. These attacks exploit vulnerabilities in authentication procedures. Unauthorized access is the primary goal, often leading to identity theft, data breaches, and other nefarious actions. Cybersecurity is under serious threat from authentication attacks, which use a variety of strategies to achieve their goal of accessing data.

#### 2.2.1. Credential Theft

The illegal acquisition of private data, including usernames, passwords, and authentication tokens, is known as credential theft. To obtain sensitive data, attackers frequently exploit flaws in IoT networks and devices, such as default login passwords, unencrypted connections, and out-of-date software. Additionally, cybercriminals may employ phishing techniques to trick customers into disclosing their login credentials. Attackers can gain unauthorized control and modify systems by intercepting user communications with legitimate services and capturing credentials as they are sent. Credential theft increases the danger of data breaches by disclosing private and operational data [21].

#### 2.2.2. Brute Force Attack

To gain access to systems without authorization, attackers employ a technique known as a brute force attack, in which they use passwords or encryption keys. To take advantage of weak passwords that users may choose, this kind of attack frequently uses automated systems. Critical infrastructure can be compromised by successful brute force assaults, which also give attackers power over IoT devices. Because smart city systems are interconnected, a breach in one component could trigger cascading failures in several other systems, potentially resulting in extensive disruptions [21].

#### 2.2.3. Replay Attack

One kind of cyber threat is replay attacks, in which a hacker records and resends legitimate data transmissions in an attempt to fool a system into carrying out commands as though they were authentic. This attack takes advantage of inadequate authentication procedures for IoT devices and networks. The cybersecurity of IoT systems in smart cities is being threatened by replay assaults. Communication between devices can be intercepted by an attacker in smart cities, and once the data are recorded, the attacker can control the system. Smart cities may be impacted by this attack in several ways. Attacks may, for instance, result in system malfunctions that affect public safety and everyday city operations by causing service interruptions. These assaults may also compromise the data integrity of IoT systems, resulting in inaccurate judgments based on misrepresented data. Security lapses like illegal access to public safety systems and infrastructure can result from replay attacks [21].

### 2.3. Protocol Attacks

Protocol attacks target the channels and communication protocols used by IoT devices to communicate with one another. Attackers can interfere with, alter, or intercept the data flow by breaching these channels, which could result in several harmful effects. Attackers use sniffing as a method to listen in on network activity. Attackers can use this technique to intercept and examine data sent back and forth between IoT devices. Data interception during transmission is one of the security hazards that may result from this [22].

### 2.4. Device Attacks

When discussing IoT authentication in smart cities, the term device attacks refers to assaults that target the IoT devices directly. These attacks do not target the network infrastructure or authentication systems. Instead, they seek to undermine the security and integrity of the individual devices. Figure 3 shows typical device attack types [14,15,23].

#### 2.4.1. Firmware Exploits

Regarding IoT device assaults in smart cities, firmware exploits pose a serious risk. Firmware is the term for the low-level software that provides the necessary operating system and functionality integrated into the hardware of IoT devices [24,25].

Figure 4 shows key information regarding firmware exploits. So, manufacturers of IoT devices should use tamper-resistant bootloader designs, frequent firmware upgrades, and secure firmware development procedures to reduce firmware attacks. To lower the possibility of successful firmware-based attacks, users should be urged to keep their IoT devices updated with the most recent firmware updates [26,27].

#### 2.4.2. Hardware Tampering

Another serious risk in the context of IoT device assaults in smart cities is hardware tampering. To undermine the security and operation of IoT devices, this kind of attack entails physically accessing and altering its hardware components [28].

Table 1 shows key information regarding hardware tampering. So, manufacturers of IoT devices should use strong supply chain security protocols, secure firmware update systems, and tamper-resistant hardware designs to reduce hardware tampering. Additionally, users should report any unusual activity or illegal access attempts and exercise caution regarding the physical security of their IoT devices [29,30].

#### 2.4.3. Malware Injection

Malware injection is a critical threat to the security of IoT devices in smart city environments. Attackers may exploit vulnerabilities in the software or firmware of IoT devices to directly inject malicious code or they may leverage social engineering tactics to trick users into installing infected software or applications on their devices. The malware introduced in this way can have a wide range of capabilities, including enabling remote control of the device, stealing sensitive data, or disrupting the normal operation of the device and the broader smart city systems it is connected to [31,32].

Malware-infected IoT devices in a smart city can have serious repercussions as they provide hackers access to vital infrastructure without authorization, interfere with necessary services, and gather private information about residents and city operations. A multi-layered strategy is needed to counter this threat, which includes putting strong device security protections in place, including runtime integrity monitoring, application whitelisting, and secure firmware upgrades. Having incident response protocols in place and regularly checking for indications of compromise are also essential for promptly identifying and containing malware attacks. Malware injection risk may also be decreased by teaching users of IoT devices security recommended practices, such as staying away from unidentified or unreliable software sources. Because malware assaults on these devices can have serious repercussions for the infrastructure and the people it serves, it is imperative to maintain the general security and resilience of IoT-based smart city systems [33,34].

#### 2.4.4. Zero-Day Vulnerabilities

When it comes to IoT security in smart city infrastructures, zero-day vulnerabilities are a serious risk. Since neither the general public nor the device maker are aware of these vulnerabilities, there are no fixes or upgrades available to address and lessen the underlying problems. As a result, attackers may actively take advantage of these unknown vulnerabilities in the hardware, firmware, or software of the IoT device to obtain unauthorized access, run malicious code, or interfere with the device’s regular operation [35].

Zero-day vulnerabilities pose a significant danger to IoT security in smart city infrastructures. There are no updates or patches available to address and mitigate the underlying issues because neither the manufacturer of the device nor the general public are aware of these vulnerabilities. Because of this, attackers could deliberately exploit these unidentified flaws in the IoT devices to gain illegal access, execute malicious code, or disrupt normal functioning [36].

Proactive security monitoring and threat intelligence collection to stay ahead of new threats, quick incident response and patch deployment capabilities to fix vulnerabilities as soon as they are found, and thorough device lifecycle management to guarantee timely firmware updates and security patches are all essential components of a multifaceted strategy to mitigate the risks of zero-day vulnerabilities. The effect of zero-day attacks may also be reduced by putting defense-in-depth techniques like device isolation and secure network segmentation into practice. The capacity to recognize and react to questionable device activity may be further improved by utilizing cutting-edge security technologies, such as machine learning-based anomaly detection [37].

Since attackers might use these unpatched vulnerabilities to create major disruptions and compromise the security and resilience of the entire system, addressing the difficulties presented by zero-day vulnerabilities is essential to safeguarding smart city infrastructure. To reduce risks and guarantee the resilience of smart city systems, an all-encompassing and proactive approach to IoT security is necessary [38].

#### 2.4.5. Denial-of-Service (DoS) Attacks

IoT devices in smart city settings are seriously threatened by DoS attacks. By interfering with IoT devices’ regular operation and accessibility, these attacks seek to prevent authorized users or systems from accessing the compromised devices [39].

In light of IoT authentication assaults in smart cities, as shown in Table 2, there are many forms that these DoS attacks can take [40,41]. A successful DoS assault against IoT devices in smart cities might have far-reaching consequences, including interfering with infrastructure monitoring and control or vital services. Strong security measures must be put in place by smart city managers and IoT device makers to lessen these threats. This entails putting in place rate-limiting mechanisms to stop resource depletion, making sure authentication procedures are safe and reliable, implementing DDoS mitigation strategies, delivering timely and secure firmware updates to fix vulnerabilities, and keeping an eye out for unusual traffic patterns to spot and react to DoS attacks [42].

To guarantee the uninterrupted and dependable functioning of vital infrastructure in the face of such disruptive threats, smart city stakeholders should improve the overall resilience and availability of IoT-enabled services by tackling the DoS attack vector. It is essential to put these thorough security measures into place to protect smart city systems’ operation and integrity against the catastrophic effects of successful denial-of-service assaults [43,44].

#### 2.4.6. Sensor Manipulation

When it comes to IoT authentication in smart cities, sensor manipulation is a form of device attack in which a hacker tries to alter the sensors or input data of IoT devices to affect the data or behavior that the devices report. Applications for smart cities, where IoT devices are frequently in charge of monitoring and managing vital infrastructure, may be especially vulnerable to this assault [45].

Figure 5 shows some common techniques used in sensor manipulation attacks. So, successful sensor manipulation attacks can have serious repercussions as they may cause inaccurate data to be sent to the control systems of the smart city. This could lead to bad choices, inappropriate use of resources, or even the erroneous activation of safety-critical systems. For instance, a traffic monitoring system’s sensors might be manipulated by an attacker to simulate traffic, causing needless detours or the initiation of emergency response protocols [46,47].

Smart city suppliers should use a variety of security measures, as shown in Figure 6, to lessen sensor manipulation attacks. Also, the overall security and resilience of an IoT-enabled infrastructure may be improved by smart city developers by tackling the vulnerabilities related to sensor manipulation [48].

#### 2.4.7. Device Impersonation

A particularly risky kind of device assault when it comes to IoT authentication in smart cities is device impersonation. To access the system without authorization, an attacker poses as a genuine IoT device. This assault can occur in many ways, like identity cloning, MitM attacks, replay attacks, and exploiting weak authentication [49].

Additionally, successful device impersonation can have serious repercussions as the attacker gains access to the IoT system, gains control of the device they have impersonated, and may be able to extend their reach to other linked devices or the larger infrastructure of smart cities. Data breaches, illegal takeovers of vital systems, and even physical disturbances or safety risks might result from this [50].

Furthermore, the total security of IoT authentication in smart city settings may be improved by using new technologies like blockchain-based identity management, conducting frequent security audits, and monitoring devices [51].

### 2.5. Data Attacks

In IoT authentication in smart cities, data attacks target the data produced, transmitted, or stored by IoT devices. These attacks aim to compromise the availability, confidentiality, or integrity of the data connected to IoT systems. Common types of data attacks, as shown in Figure 7, are discussed.

Data theft is the process of unauthorized access to and extraction of private data from IoT devices, networks, or related databases. The proliferation of IoT applications has raised the potential of data theft. A large number of IoT devices are vulnerable to exploitation and unauthorized access by attackers due to their lax security measures, which include default usernames and passwords and inadequate authentication procedures [52,53]. Compromisation of data integrity and confidentiality can come from many aspects such as Software Vulnerabilities, Network Vulnerabilities, and Hardware Trojan Attacks which all can allow adversaries to gain control over devices and access sensitive information [54].

Data manipulation refers to unauthorized modification or tampering with data produced by IoT devices. This attack presents serious hazards, particularly for vital infrastructures like smart cities, healthcare, and industrial IoT systems. Strong management systems that can adjust to the difficulties presented by data manipulation in IoT networks are required [46]. Because some of the IoT devices lack strong security measures and frequently have low computing and connectivity capabilities, they are vulnerable to data manipulation attacks. Additionally, non-consensus and failure to ensure that data are agreed upon by multiple nodes in the network leads to manipulation of data [55].

Node cloning is one type of attack that manipulates data. The attacker may send manipulated data that seems authentic by copying the identification of a legitimate node. As a result, the network’s data integrity is compromised. Also, fault packet injection is an additional form of attack. Malicious packets can be injected into the data stream by attackers. Data integrity may be compromised if this is achieved by manipulating already existing packets or by generating ones that appear authentic [56].

Additionally, the attacker may use the concept of “On–Off” attacks, in which the adversary alternates between periods of manipulation and times when they behave normally. This dynamic behavior makes data integrity detection and monitoring more difficult [46].

Data injection occurs when a hacker introduces malicious data into a system, frequently intending to alter the system’s behavior or compromise its integrity. Data injection in IoT devices can have significant consequences, particularly in smart cities where systems mostly depend on precise data. Injection attacks are a technique in which an attacker delivers malicious input (code or data) that the system interprets as authentic, resulting in unauthorized access, data breaches, or compromised data integrity. The increasing use of IoT devices that may frequently have insufficient security measures makes them susceptible to these attacks [33,57].

Different types of code injection, such as SQL injection, HTML injection, XML injection, and command injection, are categorized by the sorts of code injection attacks. Each type takes advantage of particular flaws in communication protocols and software [33]. Also, fragmentation attacks are involved, such as ChopChop attacks and ARP attacks. Targeting several layers (perception, network, middleware, and application), any of these can take advantage of flaws in the IoT architecture to obtain unauthorized access or interfere with services [57].

## 3. Authentication Mechanism

Mechanisms for authentication are essential for protecting access to data, apps, and systems. Authentication is the process of confirming user, device, or system identity. The following Figure 8 shows the basic diagram for the authentication mechanism. The user starts the registration process by entering their credentials, which the system verifies. A session is established once validation is completed, enabling safe user interaction with the system. As part of the authentication process, the system verifies the user’s permissions to ascertain which resources they are permitted to access. After that, users can request certain data, which are encrypted and sent securely. To guarantee compliance and identify any irregularities, user actions are also tracked, which enhances security overall. The system verifies the user’s decision to terminate the session, therefore removing access. The diagram highlights the crucial elements of security and data integrity in IoT contexts by emphasizing the methodical sequence of steps needed in user authentication.

Authentication solutions are essential for the security and integrity of data and devices in smart city IoT scenarios. IoT devices connect to a central authentication server during device registration, which is the first step in the process. During this step, every device is given a unique identification number and authentication credentials, such as keys or certificates. To guarantee that only registered devices may access the network, this first step connects the devices to the authentication server. The next step is user authentication. A secure login procedure is used to authenticate users. By connecting users to a database of saved credentials, this communication with the authentication server confirms their identity. The system then moves on to credential validation when a user’s identification has been verified. The authentication server verifies that a device’s or user’s credentials are current and not expired by comparing them to its records whenever they try to connect to a network or service [12].

The system puts access control mechanisms in place when validation is completed. This entails using preset policies to determine the degree of access allowed to users or devices. To regulate resource accessibility, access control systems use roles and permissions in conjunction with the authentication system. A session is established for the user or device after authentication, frequently using a session token. For a certain amount of time, users or devices can continue to access the system without having to re-authenticate, thanks to session management. The technology uses data encryption to guarantee data security. To prevent unwanted interception, data sent between users, authenticated devices, and the central server are encrypted. These communication channels are secured by protocols such as TLS, which guarantee that only parties who have been authenticated may decode the data being transferred. Furthermore, ongoing surveillance is essential to preserving security. To spot any irregularities or illegal access attempts, the system keeps an eye on user activity and device behavior. To identify suspicious activity, anomaly detection techniques frequently use ML in conjunction with authentication procedures [58].

The system contains procedures for revocation and renewal in case devices or user credentials need to be revoked, possibly as a result of loss or breach. This entails updating data, usually controlled by the authentication server, to prevent access to compromised credentials. Lastly, to make sure that authentication procedures comply with security rules and laws, frequent audits and compliance checks are carried out. Authentication events are recorded in audit logs and may be examined to evaluate security efficacy and compliance. Cities may successfully secure their IoT ecosystems while safeguarding resident data and infrastructure thanks to the general flow of operations in authentication systems for smart city IoT settings, which entail a cyclical interaction between devices, users, and the central authentication server [12,58].

### 3.1. Types of Authentication Mechanisms

Before allowing a person access to systems or data, authentication procedures are crucial for confirming their identity. As shown in Figure 9, there are a few typical categories of authentication methods.

#### 3.1.1. Something the User Knows

Information that users need to remember, such as passwords, PINs, and responses to security questions, is the main source of the “something the user knows” authentication technique. Although this approach is popular because it is simple to apply, it has serious flaws [59]. Figure 10 shows these common examples of the “something the user knows” type passwords that are usually made up of a mix of characters, numbers, and symbols. Users frequently have trouble coming up with secure passwords, which puts security at risk. PINs are short numeric codes that are frequently used in mobile devices and banking. They can be simpler to remember, but if they are not complicated enough, they can also be easy to figure out. Security questions are responses to private inquiries that may be utilized to retrieve an account. However, social media is frequently a good place to estimate or find these [60].

Several recommended practices should be used to improve the security of this type of authentication technique. First and foremost, it is essential to set difficulty criteria that motivate users to generate lengthy, intricate passwords that combine a variety of capital and lowercase letters, digits, and special characters. Because of their intricacy, passwords are far more difficult to figure out or break. Furthermore, suggesting the usage of password managers can significantly help users create and safely store these complicated passwords, lowering the possibility of weak password selections and password repetition. Passwords must be changed regularly. To further improve security, users should be encouraged to change their passwords regularly and refrain from using the same one on several sites. Also, a crucial layer of security is added by implementing 2FA. The last important factor is education and awareness by teaching users how to spot phishing efforts and stressing the need to create strong passwords so that users may better safeguard their accounts. The use of these recommended practices may greatly strengthen an organization’s security posture against unwanted access [61,62].

Research [61] emphasizes how MFA techniques must be implemented in the healthcare industry due to the serious security threats posed by conventional password-based authentication. The integration of biometric and hardware-based techniques with knowledge-based factors (such as passwords) is one of the suggested solutions; although these improve security, they also compromise user convenience and could still be subject to social engineering assaults. According to research [62], MFA and other more secure techniques are required since standard password-based authentication is becoming more and more susceptible to attacks. Along with biometrics and tokens, knowledge-based factors (passwords) are suggested solutions. These improve security but can make things more difficult to use and need to be implemented carefully to prevent new weaknesses.

#### 3.1.2. Something the User Has

To confirm the user’s identity, this authentication method uses a tangible item or gadget that the user owns. It makes it harder for unauthorized users to obtain access by adding a layer of protection on top of passwords [62]. Figure 11 shows these common examples of the “something the user has” type. Smart cards are cards with a chip installed in them to hold authentication information. They are frequently utilized in business settings and frequently call for a PIN to gain access, which improves security by fusing a user’s knowledge with something they already own. Mobile devices are capable of receiving SMS codes and creating TOTPs. Hardware tokens are little gadgets like RSA SecurID that periodically produce a unique code. The codes can be used in conjunction with passwords to offer an extra degree of protection, and they are often time-sensitive [60,62,63].

It is important to enhance security by putting best practices of this type of authentication technique. One of the best methods is MFA. Due to the requirement of various kinds of authentication, this layered approach greatly improves security and makes it more difficult for unauthorized persons to have access. Additionally, companies have to implement regular software updates for authenticating devices as the a priority. It is critical to keep these systems updated to guard against security flaws that hackers could exploit. In addition, user education is essential to security. Security breaches may be avoided by educating users about the dangers of sharing these devices and stressing the value of protecting their authentication tokens. Organizations may improve the security of their sensitive data and lower the probability of successful attacks by cultivating a culture of security awareness [62,64].

Study [60] examines immersive technology-based authentication techniques, with a focus on possession-based approaches such as smart cards, tokens, and biometric devices that improve security but also present issues with data security and user privacy. Although these techniques provide a strong defense against unwanted access, they frequently call for extra hardware and can be difficult to apply successfully in practical situations. Also, study [64] examines several possession-based authentication techniques, including SMS and hardware tokens, emphasizing that although these techniques improve security in institutions of higher learning, they have drawbacks such as device availability and attack vulnerability (e.g., SIM swapping). Although the integration of more reliable technology is emphasized in the suggested solutions, user convenience and usability may be hampered by the dependence on physical equipment.

#### 3.1.3. Something the User Is

This technique confirms a user’s identification by using distinctive physical traits. It is predicated on personal characteristics [65]. Figure 12 shows these common examples of the “something the user is” type. Fingerprint recognition examines each person’s fingertip’s distinct ridge and valley patterns. Facial recognition makes use of facial traits such as the jawline, nose shape, and eye distance. Iris recognition examines the distinct patterns in the eye’s colored region [66].

This technology, which offers improved security and user ease, is being incorporated into more and more applications. Figure 13 shows several important use cases where biometric authentication is successfully applied, such as mobile devices, banking and finance, security systems, and healthcare [67].

Study [66] examines biometric authentication techniques emphasizing approaches that use distinct physiological characteristics for safe user verification, such as facial recognition, iris scanning, and gait analysis. Strong security precautions are necessary since despite their great accuracy and resilience to common password flaws, these techniques nevertheless pose privacy problems and may be vulnerable to spoofing attacks. Also, AnswerAuth, a bimodal behavioral biometric-based authentication system, is presented in paper [67]. It uses user movements such as lifting the phone to the ear for secure access on smartphones and sliding to unlock. Despite the system’s excellent accuracy (up to 99.35 percent True Acceptance Rate) and ease of use, it may encounter difficulties due to user behavior variability and imitation attack potential, highlighting the necessity for ongoing development and adaption.

#### 3.1.4. Something the User Does

This technique verifies authenticity by examining trends in user activity. Instead of emphasizing their physical characteristics, it focuses on how people engage with technology [68]. Figure 14 shows these common examples of the “something the user does” type. Typing patterns track typing rhythm and speed to provide a distinct profile. Mouse dynamics examines the timing of clicks and movement patterns made by a user when using a mouse [69].

This method keeps an eye on how people engage with devices and systems to ensure ongoing protection. As shown in Figure 15, several important use cases highlight how successful behavioral authentication is, such as financial services, enterprise security, and fraud detection [70].

Study [69] examines behavioral biometrics-based authentication techniques, emphasizing techniques like motion patterns, touch gestures, and keyboard dynamics for ongoing smartphone user authentication. Although these techniques enable implicit authentication and provide high accuracy and user-friendliness, they have drawbacks including human behavior variability, potential mimicking weaknesses, and environmental influences. Also, study [70] examines a range of behavioral biometrics for ongoing user authentication in IoT settings, with a focus on techniques including motion patterns, touch gestures, and keystroke dynamics that make use of smartphone sensors. Although these techniques offer discreet and efficient authentication, they have drawbacks such as user behavior variability, vulnerability to imitation, and environmental influences, which call for strong security and privacy solutions.

#### 3.1.5. Somewhere the User Is

Using a user’s physical location to confirm their identity is known as geolocation-based authentication. To improve security, this can be included as an extra element in the authentication procedure [71]. Several techniques are used in geolocation-based authentication to ascertain a user’s precise position. The method uses Wi-Fi triangulation, the user’s IP address, or GPS information from mobile devices to determine the user’s position. By examining these data, the system can gain an understanding of the user’s location when they try to log in. The system could identify an attempt to log in from a different country as suspicious, for example, if the user usually accesses their account from a particular place. By detecting potentially unwanted access, the capability to evaluate the validity of login attempts according to location improves security [72]. Adhering to best practices that improve security and user experience is crucial for the successful implementation of geolocation-based authentication. First, as part of an MFA approach, geolocation should be used in conjunction with other authentication factors, such as a mobile device or a password. This multi-layered strategy greatly improves security. Prioritizing user consent and openness is also essential. To build confidence and promote system adoption, companies should explain to users why their location data are being gathered and how they are used. Additionally, adaptive security measures have to be put in place to dynamically evaluate the danger of login attempts while accounting for a variety of variables, such as user behavior, device type, and location. This flexibility enables more sophisticated access decision-making, guaranteeing seamless login for authorized users while successfully thwarting possible risks [71,72]. As shown in Table 3, common types of authentication mechanisms are displayed with some examples, advantages, and disadvantages [60,62].

As shown in Table 4, the context and necessary level of security determines which authentication method is best [59,60]. In the end, the particular security needs, user comfort, and the possible consequences of a security breach should all be taken into account when selecting an authentication method.

### 3.2. Tools and Techniques

To provide secure and reliable connectivity, authentication of IoT devices is crucial in smart cities. There are many tools and methods used, as shown in Figure 16, to achieve this goal.

#### 3.2.1. Network Access Control (NAC)

To handle the numerous security issues brought on by the growing number of connected devices, NAC is an essential part of the authentication process for IoT devices in smart cities. Each device is given a unique identification to help differentiate it from others. Only devices that have been authenticated can access the network. This is accomplished via several techniques, including token-based authentication and digital certificates, which confirm the legitimacy of IoT devices before allowing access. NAC keeps an eye on connected devices’ behavior and status all the time. To respond quickly to possible risks, this real-time monitoring aids in the detection of anomalies or compliance problems. NAC can remove a device from the network if it shows suspicious activity to stop more dangers. Access control models like RBAC and ABAC specify permissions based on user roles and various attributes, respectively; these techniques enable devices to securely authenticate their identities. Access rights must be continuously monitored to allow for real-time permission updates when circumstances alter [2,73].

#### 3.2.2. Edge Computing Authentication

Edge computing stores data and brings computing next to where it is needed. Edge computing plays a role in authenticating IoT devices in smart cities. Edge computing architectures implement mutual authentication protocols, where the identification of the edge server and the IoT device is verified before a connection is established. This verification helps prevent unauthorized access and guarantees that the network can only be accessed by authorized devices. The data transmission between the device and the server is encrypted using the session key, ensuring the confidentiality and integrity of the communication and its contents [74,75]. In addition, edge computing uses lightweight authentication protocols to reduce computational overhead [75]. Edge computing is flexible because it supports dynamic authentication based on devices and their behavior [76].

#### 3.2.3. Zero Trust Architecture

ZTA is a robust framework for authenticating IoT devices in smart cities. This approach works on the principle that by default, no device or user can be trusted, no matter where it is. Every request for access from inside or outside the network must be verified and authorized. Each device is assigned a unique identity, often through digital certificates. This guarantees that the network can only be used by authorized devices. Also, constant monitoring of all devices is required to check their behavior. A baseline of typical device behavior is established. Any deviation from this baseline triggers an alert and additional authentication is requested and that ensures security. This is important in the context of smart cities where many IoT devices interact across different networks [77,78].

#### 3.2.4. Cryptographic Algorithms

In smart cities, cryptographic algorithms are crucial for protecting communications and guaranteeing the integrity and confidentiality of information sent between IoT devices. The main cryptographic methods used for IoT authentication are hash functions, digital signatures, symmetric and asymmetric encryption algorithms, and key exchange algorithms. Symmetric encryption algorithms use a single key for both encryption and decryption processes, making them effective for environments with limited processing power. This type uses algorithms such as AES, DES, and 3DES. As for asymmetric encryption algorithms, they use two encryption keys. A private key for decryption and a public key for encryption are available. This type uses algorithms such as RSA and ECC. In hash functions, input data are transformed into a fixed-length character string, usually a digest that reflects the contents. They are necessary to guarantee the integrity of the data. This type uses algorithms such as Secure Hash Algorithm 256-bit (SHA-256) and HMAC. As for digital signatures, they use asymmetric encryption to ensure authenticity and prevent eavesdropping by unauthorized persons. This type provides a guarantee of the origin and integrity of the message. Finally, key exchange algorithms allow devices to securely share cryptographic keys. This type uses algorithms such as Diffie–Hellman key exchange [79,80,81].

#### 3.2.5. Public Key Infrastructure

To ensure secure authentication of the IoT in smart cities, it is necessary to use PKI. The reason is that by enabling secure key management and enabling devices to use reliable digital certificates for mutual authentication, the PKI improves the security of communications. It consists of four key components, which are digital certificates, CAs, RAs, and CRLs. The most important benefits that PKI provides when used are the following: it enhances security by providing strong authentication mechanisms; it makes it capable of expanding and accommodating new devices on the Internet of Things without compromising security; it supports many different standards and protocols that make or allow devices to communicate securely and effectively; it provides automated management. However, there are some challenges and considerations that must be taken into account when implementing PKI, which are resource constraints, trust management, and lifecycle management. Likewise, to improve PKI for Internet of Things applications, recent research has concentrated on creating lightweight protocols that preserve security while lowering overhead. For example, the AutoPKI framework reduces manual involvement and enhances scalability for IoT installations by automating credential changes and trust transfer. Furthermore, research has looked into combining PKI and blockchain technologies to improve trust in IoT settings [58,82].

#### 3.2.6. Biometric Authentication

The need for biometric authentication as a security measure for IoT devices in smart cities is becoming more widely acknowledged. This method takes advantage of unique physical characteristics, such as face recognition, fingerprints, and iris patterns, to improve security and simplify user access. Some technologies are used in biometric authentication, which are fingerprint scanners, facial recognition systems, and iris recognition. One of the most important things that biometric authentication provides in IoT devices in smart cities is enhancing security because it offers more protection than conventional password-based solutions. Because physical characteristics are unique to each person, it is impossible to copy or steal them, which makes illegal access more difficult. However, many challenges and limitations are faced when applying or implementing biometric authentication in IoT devices in smart cities. When biometric information is hacked, it cannot be changed like passwords. To prevent hacking of biometric data templates, there must be strong encryption and secure storage technologies. Also, the application of biometric authentication produces complexity in the system, and it may be difficult to integrate biometric systems with the existing IoT infrastructure. Many aspects of biometric authentication have been examined in recent studies of IoT environments in smart cities. To improve and increase security, the focus is on combining biometric information with cryptographic methods. There is also a focus on wearable biometrics, which work by continuously monitoring vital characteristics. These sensors can improve and increase security and provide authentication in real time. Also, combining biometric authentication with additional authentication methods, such as passwords, is called MFA [75,83,84].

#### 3.2.7. Multi-Factor Authentication

One of the critical security features for IoT authentication in smart cities is MFA, which provides an extra layer of security by combining two or more independent authentication credentials. This can significantly reduce unauthorized access even if one of these factors is compromised. Some challenges and considerations still exist despite the great benefits provided by MFA, which are the user’s inconvenience when applying complex MFA procedures and that they are very expensive when applied. Many aspects of MFA design have been examined in recent studies of IoT environments in smart cities. To improve the security of MFA systems, new frameworks are being created that use decentralized technologies such as blockchain. Researchers are also looking for lightweight MFA protocols that maintain security without overtaxing device capabilities due to the limited resources in many IoT devices. Also, one of the reasons why MFA systems are increasingly integrating biometric authentication is because it enables seamless access to the service, and this integration increases security and improves user convenience [61,85,86].

#### 3.2.8. Blockchain Technology

In smart cities, blockchain technology has become a game changer in the IoT authentication space, solving security issues due to the growth of connected devices. Making use of blockchain’s decentralized and unchangeable characteristics lowers the danger of fraud, improves trust and security in device authentication, and improves the reliability of IoT systems. Among the most prominent benefits that blockchain technology provides for IoT authentication are decentralization, strong security for data transactions, data integrity, smart contracts, and scalability. Several applications and frameworks have been investigated to integrate blockchain technology with IoT authentication in recent studies. To guarantee that only devices with authorization may access or connect to the network, blockchain technology has been proposed to authenticate and register IoT devices in smart cities. Also, the combination of blockchain technology with edge and fog computing models enhances the efficiency of IoT systems. Although blockchain technology offers significant advantages, it faces significant challenges, such as interoperability and energy consumption [75,87,88].

These technologies are compared, as shown in Table 5, in terms of scalability, security, and latency. The trade-offs associated with choosing various technologies depending on certain needs and situations are highlighted by this comparison.

## 4. Related Work

### 4.1. Methodology

This methodology delineates the procedures for performing a systematic literature review focused on authentication mechanisms for IoT devices within smart cities. The objective is to consolidate current research, pinpoint trends, recognize gaps, and address challenges. This study aims to investigate the existing authentication mechanisms employed for IoT devices within smart cities and assess their effectiveness in maintaining security. In our research paper, we concentrate on studies that look at the examination of authentication methods tailored for IoT devices within the framework of smart city environments. Peer-reviewed publications released between 2019 to the present are the main focus of the selection criteria. Articles that do not fit these requirements are disqualified, and only research that explicitly addresses IoT authentication in smart cities is taken into consideration. We begin our academic search by utilizing the Google Scholar. These platforms are selected because they have wide and great amounts of peer-reviewed cybersecurity and IoT literature. Specifically, the search is conducted using terms like IoT authentication, smart city security, authentication mechanisms, and cybersecurity threats in IoT. This focused strategy makes sure that relevant information is retrieved.

### 4.2. Selection of Research Papers for Review

To guarantee the topic’s comprehensiveness, the research articles are chosen through a methodical and exacting procedure of assessing the literature on IoT device authentication in smart cities. Databases that provide academic research are first searched using keywords related to the research topic. Papers are evaluated in terms of their relevance and contributions to the field. The goal is to collect sufficient literature to gather enough results to identify and fill gaps in the field. In this research review, the PRISMA 2020 flowchart is used as shown in Figure 17, providing a comprehensive framework for reporting this systematic review. We first start by searching through Google Scholar for research related to IoT devices and the cyber issues that we may face when using them in smart cities. In addition to the authentication mechanisms used, blockchain regulation, edge computing, presenting fresh authentication frameworks, and techniques designed for Internet of Things environments are considered. To guarantee that the included literature directly addresses security protocols and methodologies applicable to IoT devices in smart cities, each study needs to be relevant to authentication processes. Priority is given to studies that offer empirical assessments of current authentication frameworks or techniques to guarantee that useful insights are included. Since they advance our knowledge of the problems and solutions in the industry, articles that offer thorough security frameworks for IoT in smart cities are also highlighted. Based on their innovative contributions to the field, selected research is included.

Initially, the number of initial research papers was 223 according to the specified keywords. A revised collection of research was obtained by eliminating duplicates. A total of 101 papers that did not fit the inclusion criteria were excluded after preliminary screening of titles and abstracts was performed to weed out research that was not relevant, leaving 122 papers. The remaining 122 papers underwent full-text evaluations to ascertain their eligibility and pertinence under the inclusion criteria. In the end, 47 publications that made significant contributions to the subjects of IoT security and authentication in smart cities were included.

By synthesizing findings from recent studies, next sections analyze previous studies regarding the integration of IoT in smart cities, highlighting the challenges facing authentication and security mechanisms in urban environments and seeks to provide an overall understanding of the important aspects related to the integration of IoT in a smart city environment, focusing on security issues related to authentication. In addition, Table 6 comprehensively summarizes the related studies.

### 4.3. Types of Attacks

#### 4.3.1. Replay Attacks

Ali et al. [4] emphasize device authentication, access control, and a thorough analysis of IoT security procedures. They cover a range of authentication strategies, from conventional password-based systems to more sophisticated approaches like biometric and token-based authentication. The authors stress the necessity for lightweight authentication solutions that take into account the resource limitations of IoT devices while highlighting the efficacy of these techniques in reducing common dangers like illegal access and data breaches. The absence of a defined architecture for building IoT applications in the paper makes it challenging to guarantee uniform security across a range of use cases. Additionally, it points out a vacuum in the literature’s investigation of support layer security vulnerabilities, which have not been fully examined.

Nandy et al. [89] present the security techniques for IoT authentication that highlight the significance of safe authentication processes given the diversity and vulnerability of devices in IoT networks. In addition to examining existing security approaches and suggesting improvements, the study describes a variety of attacks against IoT authentication, including masquerading attacks, MitM attacks, DoS attacks, and forgery attacks. The main shortcoming of this study is that, while it points out several unresolved issues and bottlenecks related to IoT security, it does not provide comprehensive empirical support for the proposed fixes or a comprehensive assessment of their success in practical situations.

#### 4.3.2. MitM Attacks

Xihua and Goyal [5] present privacy and security concerns in smart cities, focusing on how blockchain and IoT technologies can be used to solve these problems. They draw attention to vulnerabilities in smart city infrastructure that could lead to data leaks and unauthorized access to private data. The paper’s shortcomings are that while it identifies several security issues and proposes blockchain as a solution, it may not thoroughly examine the implementation difficulties or practical application of blockchain in different smart city scenarios, nor does it delve into the potential trade-offs between security and usability.

To protect sensitive data and device integrity, Kamarudin et al. [58] provide a thorough analysis of authentication techniques for IoT, emphasizing the critical need for robust security measures. They look into several authentication techniques and highlight their benefits, drawbacks, and unique problems brought up by IoT devices with restricted resources. To thoroughly evaluate existing protocols, identify trends, and draw attention to security concerns, this study employs a multi-criteria classification method. By combining knowledge from recent studies and offering useful recommendations for practitioners and policymakers, the authors hope to lay the groundwork for future research and improve knowledge of the intricacies of IoT authentication and its security implications in this rapidly evolving field.

#### 4.3.3. DoS Attacks

Syed et al. [10] offer a thorough explanation of IoT in smart cities, delving into its essential elements, enabling technologies, common uses, and deployment difficulties. They seek to demonstrate how IoT may improve urban living by promoting sustainability and better services. This research has some flaws that may lead to the absence of details in some areas and the possibility of bias when discussing difficulties without a thorough expert consensus, which might restrict the findings’ generalisability across various smart city implementations.

Gayathri et al. [18] discuss and focus on DDoS and fake data injection attacks and look at a framework for enhancing the security of IoT devices. The proposed system combines MTD techniques for mitigation with ACL and SNMP for attack detection. Tests demonstrate a considerable decrease in attack delay and frequency. This paper’s major focus on DDoS and fake data injection attacks, however, may overlook other new dangers in the rapidly evolving IoT ecosystem that may require additional security measures beyond those recommended.

Pereira et al. [11] discuss the difficulties encountered by IoT devices with limited resources, with a particular emphasis on energy and communication as crucial success elements for upcoming deployments. To overcome these obstacles, they investigate several technologies, including EH, WPT, and backscatter transmission. The paper’s shortcomings, however, include a dearth of empirical evidence to back up the suggested technologies and procedures as well as a lack of investigation into practical implementation problems like security and interoperability with current systems, which could make it more difficult to put the solutions discussed into practice.

#### 4.3.4. Physical Access Attacks

Al-Turjman et al. [1] discuss how IoT communications in smart cities raise privacy and security issues. They discuss the improvements in services and quality of life in urban environments from the perspective of collecting private and sensitive data. The main uses of smart cities are explained, the concerns about security that arise in their design are discussed, current solutions to these problems are examined, and future lines of investigation are proposed for improvement. They also highlight the importance of considering security issues when designing smart technologies and propose a secure IoT construction for smart cities. One of the paper’s many shortcomings is its limited list of suggested security and privacy fixes, which could not cover all cutting-edge methods or technologies.

### 4.4. Types of Authentication

#### 4.4.1. Password-Based Authentication

In a paper by Majid [15], the integration of IoT technology in smart cities is examined, which highlights the significant security concerns that arise from this integration. It examines several applications of smart cities, such as traffic control and healthcare, while identifying IoT system flaws that malevolent actors can exploit. In addition to emphasizing the necessity of strong security measures, the report provides research opportunities and possible solutions. However, one of the paper’s limitations is that it does not fully cover security and privacy solutions for smart systems at all levels or provide an in-depth analysis of how well suited the security mechanisms in place are to fend off sophisticated attacks.

Wang et al. [90] focus on both existing password systems and new biometric systems, and the article provides a comprehensive analysis of the vulnerabilities of user authentication systems and their responses. In addition to providing evaluation criteria for defense measures, typical attacks are classified according to elements such as attacker knowledge and attack type. The lack of effective defense mechanisms against a wide range of potential attacks is a limitation of the paper; many existing approaches primarily address specific types of attacks and do not often adapt to new threats, emphasizing the need for more flexible security solutions.

#### 4.4.2. Biometric Authentication

Ng et al. [83] discuss how biometric information is used in smart city projects, stressing how crucial it is for monitoring, identification, and authentication. To examine the processes of data origination and recognition, they divide biometric features into physiological and behavioral categories. The acceptability of data collecting and the renewability of biometric traits are the main topics of this study, which emphasizes the importance of digital ethics. In addition to arguing that a combination of renewable and nonrenewable traits can increase identification accuracy and reduce the likelihood of identity theft, the authors argue the need to obtain individuals’ explicit agreement before collecting biometric data to protect privacy. To support the development of smart cities, they ultimately call for more ethical procedures in the gathering of biometric data.

To enhance security in IoT environments, Yang et al. [84] discuss the integration of encryption and biometric authentication systems. The proposed method categorizes various biometric attributes, examines existing systems, and identifies problems and potential solutions related to biometric applications in IoT. This paper’s main flaw is that, although providing a broad overview, it does not go into great detail on every historical advancement and real-world use of biometric systems, which may limit knowledge of their true usefulness and efficacy.

#### 4.4.3. Blockchain-Based Authentication

Khalil et al. [3] compare centralized and blockchain-based architectures to examine the methods of authentication for IoT devices in smart cities. The study examines the body of research on authentication procedures, describes security issues, and discusses current developments and potential avenues for further study. Although the research highlights the need for security procedures and identifies several open research challenges, it may not provide comprehensive solutions to address the technical issues involved in successfully implementing these authentication architectures in real-world settings.

Given the impending widespread use of IoT made possible by 5G networks, Ferreira et al. [75] investigate how blockchain technology might be included to enhance the IoT device authentication and registration processes within smart city applications. They highlight the challenges with security and scalability associated with non-standard IoT devices, particularly those situated in remote urban locations. To ensure reliable data transmission despite potential risks from old firmware, the study proposes a methodology that uses API gateways that use smart contracts for message signing, identification, and validation. Their strategy uses the edge and fog computing paradigms to create a decentralized security framework that increases efficiency and trust in the handling of IoT data in smart cities.

#### 4.4.4. PKI

Wu et al. [74] propose an anonymous authentication mechanism for smart home environments that uses edge computing and PUFs to enhance user-smart device communication security. By using edge computing for increased data processing efficiency, they address the drawbacks of traditional cloud computing, which struggles to meet real-time customer requirements. PUFs help stop data leaks and assaults, and the protocol ensures mutual authentication and the creation of a secure session key. Using various models and analyses, the authors verify the protocol’s security and demonstrate improved security and performance when compared to existing techniques in the field.

An inventive lightweight cryptosystem designed for IoT applications in smart city environments is presented by Hazzaa et al. [79]. They address the challenge of preserving data security while taking into account the limited computing power of IoT devices. By using a dual XOR S-box technique and reducing the number of encryption rounds from 10 to 9, the proposed algorithm enhances the AES. This modification aims to preserve security standards while reducing execution time and energy consumption by about 33 percent. The authors demonstrate through testing with various audio samples that the proposed method successfully strikes a compromise between security and performance; as a result, the suggested method is appropriate for real-time multimedia applications in environments with restricted resources.

#### 4.4.5. MFA

Suleski et al. [61], discuss and examine MFA solutions in IoHT, focusing on evaluating current technologies, identifying security requirements, and addressing the problems and vulnerabilities associated with healthcare authentication. The authors stress the value of robust MFA procedures to protect private medical information, particularly in the context of increasing cyber threats. This study’s main drawback is that it highlights shortcomings in existing MFA solutions, particularly about passwordless techniques in IoHT, without offering a thorough analysis of implementations or in-depth case studies that could offer insightful information for improving authentication security in healthcare settings.

Bamashmos et al. [86] discuss and present a 2L-MFA architecture that uses decentralized blockchain technology to increase the security of IoT users and devices. Its first layer focuses on IoT device verification using secret keys, location, and PUFs, while its second layer deals with user authentication using identity, passwords, and biometrics such as iris and finger vein recognition. The main flaw of the suggested approach is its dependence on the intricacy of PUFs and the need for lightweight algorithms, which could make it difficult to deploy on IoT devices with constrained resources and might not fully address all possible security threats in the rapidly developing field of IoT threats.

Bulat and Ogiela [9] discuss context-based personal authentication techniques that have been specifically created for the IoT. They discuss the particular security issues faced by low-power devices in IoT settings and propose an authentication technique that uses several independent variables, such as contextual data and user attributes. The goal of this technique is to improve authentication reliability while maintaining usability in resource-constrained environments. The authors emphasize the importance of privacy, sensitive data security, and ethical issues when creating such authentication systems. The drawbacks of this paper is the absence of comparison with current authentication techniques that includes performance measurements and benchmarks. This can illustrate the benefits of the suggested strategy.

#### 4.4.6. Cryptographic Algorithms

The performance consequences of several encryption algorithms for resource-constrained devices that are often used in IoT communications—especially in smart homes—are examined by Silva et al. [81]. Using two transport protocols, HTTP and MQTT, the researchers conduct trials with six embedded devices and assess variables such as power consumption, message latency, and extra message length. They focus on lightweight cryptographic algorithms, specifically AES and ChaCha20, to address security concerns in IoT environments. The results highlight the need for efficient security measures in low-power devices by demonstrating significant differences in message delays and additional bytes based on the chosen cryptographic techniques. The goal of this study is to improve our understanding of how cryptographic choices affect performance in typical smart home setups.

In a paper by Alatawi [80], a discussion is presented of growing security concerns over the proliferation of IoT devices, many of which have restricted memory and processing power. A HAC approach is proposed to enhance IoT data transmission network authentication methods. This method combines a hybrid encryption technique that combines the RSA and AES algorithms with lightweight cryptographic techniques that specifically use the EX-OR operation and a hashing function. By addressing authentication process flaws, the authors aim to enhance the overall security of IoT applications and ensure privacy in smart city infrastructures.

### 4.5. Security Frameworks

#### 4.5.1. ZTA

Liu et al. [77] analyze the zero-trust security paradigm, emphasizing its significance for network security, particularly IoT. To assess the current state of research on zero trust, a bibliometric analysis is conducted, looking at shortcomings, practical uses, and implementation challenges in IoT environments. This paper describes the key components and technologies of a zero-trust architecture, such as MSG, IAM, and software-defined perimeter. Additionally, the paper highlights new research trends and possible future paths, as well as the importance of applying zero-trust principles to handle security threats inherent in IoT systems.

In the framework of zero-trust security for the IoT, Bast and Yeh [78] look into new authentication technologies. They highlight the unique security challenges brought on by the resource-constrained and networked nature of IoT devices, which makes conventional protection inadequate. This study emphasizes the importance of robust authentication methods, such as blockchain technology, mutual authentication, and lightweight cryptography, to provide safe communication and access management. The approach, which is central to the zero trust idea, states that continuous identity verification is crucial to lowering cyber threats. They also discuss the challenges of implementing these technologies in various contexts and offer ideas for future research paths to improve IoT resilience and security.

Andrade et al. [21] look at how integrating IoT technologies into smart cities leads to serious cybersecurity challenges. They highlight how smart cities address significant cybersecurity issues and enhance public services across the board by leveraging IoT, Big Data, and cloud computing. After conducting a thorough literature analysis, the authors present a cybersecurity maturity model for assessing the cybersecurity maturity of IoT solutions in urban environments. This study emphasizes how important it is for city administrators to put robust cybersecurity measures in place to lower risks, boost public trust, and keep smart city networks resilient to growing threats.

As billions of objects are connected to networks, Ragothaman et al. [73] conduct a thorough analysis of access control in IoT, emphasizing its importance in preserving security and privacy. They highlight the unique difficulties posed by the diversity of IoT devices, including resource constraints and shifting environmental conditions. Important access control requirements, architectural frameworks, and policies are examined in this study, emphasizing how often existing solutions fall short of these requirements. The authors discuss the drawbacks of different access control models, such as attribute-based, role-based, and discretionary access control, as well as the research issues and future directions required to develop efficient access control systems appropriate for IoT applications.

#### 4.5.2. IDS

Abosata et al. [17] discuss and provide a detailed analysis of the security difficulties faced by the IIoT and IoT. The Perception, Network, Processing, and Application layers of the IoT architecture are used to categorize various threats and security measures. Current security measures, including IDS, cryptography, and communications protocols, are examined by the writers. New techniques for assessing these mechanisms are also discussed. However, a major flaw is that the paper mostly focuses on recent surveys and might not fully address the rapidly evolving landscape of IoT security. Moreover, it does not include fresh dangers and solutions that have emerged since its publication.

El-Sofany et al. [19] discuss and look at a novel ML-based security model that aims to strengthen the security of IoT systems and address the increasing vulnerabilities as IoT applications spread across many industries. The work offers a thorough analysis of contemporary technologies, security concerns, and the advantages of ML to manage and lessen security threats on its own while achieving high performance and accuracy standards. However, a drawback of the research is that it acknowledges the challenges of applying ML in IoT networks, particularly due to the limited processing power and energy of IoT devices and the challenges posed by the diverse and continuous streams of real-time data generated by these systems.

### 4.6. Emerging Technologies

A paper by El-Jaouhari [7] illustrates the importance of strong authentication frameworks while discussing developments in approaches to authenticating IoT devices to protect interconnected systems. Based on the existing body of the paper, the authors categorize different authentication procedures, including new lightweight protocols designed for resource-constrained devices and more traditional techniques. To improve authentication procedures, the study also looks at new developments, such as the technologies of blockchain and edge computing. The overall goal of the research is to provide a comprehensive classification of IoT device authentication while pointing out any gaps or difficulties that require further research. However, the paper lacks case studies or real-world application examples that illustrate authentication techniques in action.

Dave et al. [76] talk about how edge computing might improve data transmission speed, security, and efficiency in three important areas: the IoT, smart cities, and healthcare. They emphasize that by decreasing latency and handling data nearer to their source and bandwidth consumption, edge computing overcomes the drawbacks of traditional cloud computing. These systems improve patient outcomes in healthcare by enabling real-time monitoring and feedback from medical devices. Edge computing improves resource optimization and energy management in smart cities by analyzing data in real time from several IoT devices. To fully realize the potential of edge computing, this article highlights the need for more research to overcome current restrictions, such as security issues and the implementation of 5G networks.

Kuzlu et al. [16] discuss and examine how AI enhances cybersecurity for IoT, with a focus on threat identification and the challenges posed by hostile AI utilized by cybercriminals. It explores AI approaches used in cybersecurity, such as ML, and looks at various attack strategies targeted at IoT devices. The paper’s reliance on already published works rather than unique empirical research is one of its main limitations.

Dritsas and Trigka [88] examine the integration of blockchain, IoT, and ML into smart cities, with a focus on how these technologies might boost data analytics, security, and operational efficiency to improve urban management. While addressing challenges like scalability, privacy, and ethics, this study scouts the revolutionary potential of ML-powered blockchain-IoT ecosystems in creating robust and sustainable urban infrastructure. The study’s reliance on previously published works and lack of empirical case studies, however, are serious limitations that may affect the applicability of its conclusions and insights for actual smart city deployments.

To identify trends, important studies, and potential directions for future study, Rejeb et al. [87] discuss the use of blockchain technology in smart cities. This highlights how quickly research in this area is growing and emphasizes how blockchain technology might revolutionize urban infrastructure by enhancing security, efficiency, and transparency. However, a major drawback of this study is its focus on English-only publications that are listed in the Scopus database, which may overlook relevant research in other languages or databases, potentially skewing the results and limiting the review’s comprehensiveness.

A paper by Badidi [13] discusses how blockchain technology and edge AI can be combined to create smart, sustainable cities, highlighting how they can improve energy management and urban mobility. It examines how these technologies might offer creative answers while highlighting contemporary issues that smart cities face, like traffic jams and energy waste. The study does, however, note certain drawbacks, such as the still-infancy state of research on edge AI and blockchain integration, as well as possible problems with privacy, scalability, and the requirement for strong data management frameworks to guarantee the effective implementation of those technologies.

With a focus on security, privacy, and trust issues within a three-layer IoT framework, Adam et al. [22] provide a thorough analysis of the major concerns surrounding the IoT. The pervasiveness of IoT technology and its revolutionary impacts are examined in this work, along with the significant risks and vulnerabilities brought on by its enormous growth. To safeguard sensitive data and defend IoT systems from threats, this study looks at recent developments and approaches that address security requirements and countermeasures. To help researchers and practitioners manage the complexities of IoT security and privacy, the authors also identify unresolved issues in the security landscape and provide potential solutions. They achieve this by attempting to make connections between fundamental and advanced themes.

Ismagilova et al. [20] discuss and explore the complex problems of security in smart city settings. The paper gathers the body of knowledge about information security threats, mobile device privacy concerns, and operational risks faced by smart city infrastructures. To address these issues and guide further research, the authors develop a paradigm for smart city interaction. The paper’s focus on synthesizing existing research without providing actual evidence or case studies to support the suggested framework, however, is a major flaw that might limit its applicability in real-world scenarios.

### 4.7. General Challenges and Solutions

Whaiduzzaman et al. [2] explore the features, applications, and integration of advanced technologies such as blockchain and ML to improve smart city operations. They achieve this by providing an overview of emerging technologies in the context of IoT-based smart cities. Besides addressing several security issues and concerns, the paper promotes a more robust smart city architecture. However, one of the drawbacks mentioned is the focus on specific issues and uses rather than a close examination of the broader impacts of IoT technologies on infrastructure and social dynamics, which may have an impact on the adoption and deployment of these smart city technologies.

To increase the long-term maintainability and scalability of IoT deployments, Höglund et al. [82] propose a lightweight protocol that automates control transmission between service providers. They address the problems with traditional PKI in IoT environments, which often suffer from high manual overhead and vendor lock-in. By using formal verification techniques, the AutoPKI protocol offers excellent security while minimizing the requirement for human input during trust exchanges. Through a prototype implementation, the authors demonstrate the feasibility of their approach and assess its performance on key security requirements, highlighting its potential to enable effective and secure control of IoT devices across many service providers.

Khalid et al. [91] present a lightweight, decentralized blockchain-based authentication and access control system designed for IoT applications. By leveraging blockchain’s fault tolerance, tamper proofness, and reliability, this system seeks to improve the security of IoT devices that frequently handle sensitive data. The drawback of this mechanism is that it may lead to increased communication costs, which may lead to higher power usage, particularly for IoT devices with limited resources. Moreover, although the proposed mechanism offers greater scalability than centralized approaches, it still suffers from communication control issues between groups of different devices and may require multiple message exchanges for authentication, which may make things even more difficult.

Thakare and Kim [8] address the issues of high resource consumption and latency associated with current protocols by presenting an innovative, secure, and lightweight authentication technique tailored to IoT contexts. Considering the IoT devices’ low processing and power capabilities, it highlights the need for efficient authentication techniques. The proposed method improves authentication procedures while reducing computational, communication, and storage overheads by using ECC and one-way hash functions. The authors highlight the implementation results of the scheme, which show significant increases in operational efficiency when compared to existing methods and demonstrate its effectiveness against a variety of security risks. In this paper, the user experience component of the authentication process receives little attention. It would be useful to know how the proposed scheme impacts end user interactions with IoT devices.

Omrany et al. [12] discuss the application of IoT in smart cities, emphasizing important research topics, difficulties, and potential paths forward. To determine four main research areas, they perform a hybrid systematic analysis of 843 papers published between 2010 and 2022. The paper’s shortcomings, however, include its dependence on a single database (Web of Science), which might not contain all pertinent material, and the possibility of knowledge fragmentation due to the multidisciplinary nature of IoT and smart city research. Furthermore, the field’s quick developments could surpass the thorough review’s pace, leaving new themes unaddressed.

With an emphasis on the status of cybersecurity and cyber forensics today, Kim et al. [14] discuss and examine how new technologies like cloud computing, IoT, and AI are integrated into smart cities. The paper examines 154 studies published between 2015 and 2022, emphasizing important research topics and difficulties, with a focus on ongoing studies on IoT and smart homes. The paper’s narrow focus on cyber forensics, which is less studied than cybersecurity, is a major drawback, though, and might compromise all-encompassing security plans for smart city infrastructures.

Altulaihan et al. [6] review the cybersecurity risks associated with the IoT, as well as defenses and mitigation strategies. They focus on the rapid rise of IoT devices and the risks they present due to inadequate information security. By examining the relevant literature, the authors classify different cybersecurity risks according to the levels of the IoT architecture and review typical mitigation techniques and countermeasures. Their goal is to provide an overall understanding of the specific risks of the IoT and offer suggestions for future lines of investigation. The work also highlights the importance of protecting data and services in IoT contexts, describes the study methodology, and highlights important conclusions about specific types of threats. However, one of the shortcomings in this paper is that comprehensive instructions on how to use the recommended cybersecurity countermeasures in actual IoT applications are not included in the study.

### 4.8. Emerging Research Directions

Singh and Singh [92] offer a thorough analysis of authentication techniques for IoT devices, highlighting how important they are to creating secure communications and safeguarding private information. The research covers a range of authentication strategies, from more conventional to more recent blockchain-based systems, emphasizing how well they work to manage access privileges and confirm user identities. The deployment of strong authentication systems may be hampered by the resource restrictions of IoT devices, for example, which are acknowledged as key limitations in the study. The paper also highlights the possibility of vulnerabilities as a result of many traditional approaches failing to sufficiently address the particular difficulties presented by the wide and diverse IoT landscape. Overall, even though the paper points out encouraging developments in authentication, it urges more study to create effective and scalable solutions that are suited to the resource-constrained nature of the IoT.

With an emphasis on the difficulties in protecting smart home IoT devices from different cyber threats, Kumar et al. [93] offer a simple authentication system. They highlight the use of asymmetric key cryptography for user and device authentication, outlining a procedure that uses authentication coupons and offline public–private key pair generation to guarantee a safe connection. Although the suggested techniques successfully thwart replay and MitM attacks, the study notes drawbacks associated with the processing limitations of low-power Internet of Things devices. Even while using cryptographic approaches improves security, there may still be issues with resource consumption and the requirement for strong user management. The paper’s overall message emphasizes the necessity of striking a balance between security protocols and IoT device functionality.

To improve security and scalability, Mukhandi et al. [94] highlight a consensus authentication method in its blockchain-based identity management strategy for IoT devices. The work describes how IoT nodes use signed IDs that are cross-checked against a tamper-proof blockchain ledger to authenticate one another. By reducing the dangers of bottlenecks and single points of failure that come with centralized authentication methods, this decentralized approach makes the system more robust as the number of devices grows. The computational overhead of blockchain transactions, which might strain IoT devices with limited resources, is one of the drawbacks mentioned in the paper. Even though the suggested system is more scalable and has fewer authentication delays than conventional techniques, it still has to be further optimized because it depends on consensus procedures, which could cause latency in situations with high demand.

### 4.9. Benchmark Studies on Authentication

To increase security and lessen vulnerabilities in devices with limited resources, Ahmed and Ahmed [85] discuss a reliable multifactor authentication system for IoT that uses different hash function combinations. To address issues with temporal synchronization in traditional authentication methods and guarantee data integrity and validity, this study presents a novel mechanism called TEOTP. This research studies specific authentication techniques without fully addressing the broader cybersecurity requirements for IoT networks or the potential challenges of implementing these solutions in various IoT contexts with disparate device capabilities.

Rao and Deebak [95] present the security concerns in smart businesses and cities with a focus on IoT applications. The applications, technologies, and challenges arising from vulnerabilities in networked systems are examined. The paper lacks standard technologies and protocols for IoT applications, making it difficult to manage security, privacy, and performance efficiency, as well as the large computational resource requirements and battery life considerations of the devices, which constitute a major limitation of the paper.

**Table 6 sensors-25-01649-t006:** Related studies.

Reference	Year	Methodology	Technology	Sector/Application	Open Issues	Limitations
[1]	2022	Mixed	IoT, Blockchain, Cryptography, Biometric Systems, and ML	Healthcare, Transportation, Smart Home	Privacy, scalability	Limited scope
[2]	2022	Mixed	IoT, ML, Cloud and Edge Computing Blockchain	Healthcare, Transportation, Smart Home	Privacy, Scalability, and Energy efficiency	Limited practical implementation insights
[5]	2022	Mixed	IoT and Blockchain	Healthcare, Smart Cities, Education,	Privacy, scalability, interoperability	Data privacy
[3]	2022	Mixed	Blockchain, AI, Cloud Computing, Fog Computing	Smart Cities, CPSs, and IoT	Authentication mechanisms, scalability,	Focus on Specific Architectures
[6]	2022	Mixed	IoT and Protocols such as MQTT	Healthcare, Smart Cities	Privacy, scalability, interoperability	Focus on common protocols
[4]	2019	Qualitative	IoT	Healthcare, Transportation, Smart Cities	Authentication, access control, scalability	Coverage of recent advancements, lack of empirical data
[7]	2023	Mixed	IoT, Blockchain, ML, AI	Healthcare, Smart Cities	Resource constraints, threats	Rapid evolution, generalization
[8]	2021	Quantitative	ECC, Cryptographic Hash Functions	IoT, Cloud Computing	High costs, scalability, privacy	Limited formal security analysis
[9]	2023	Mixed	IoT, Biometrics, cryptography	IoT	Privacy, spoofing	Refinement requirements
[10]	2021	Mixed	IoT, AI, Cloud and Edge computing	Healthcare, Transportation, Smart Cities	Privacy, scalability, interoperability	Generalization of findings
[11]	2020	Mixed	Wireless Power Transfer, Energy Harvesting	IoT, Smart Cities, Healthcare	Privacy, energy efficiency, data management	Rapid technological changes
[12]	2024	Mixed	IoT, Blockchain	Smart City	Privacy, scalability, interoperability	Fragmentation of literature
[13]	2022	Qualitative	Edge, AI, Blockchain	Smart Mobility and Energy	Privacy, scalability	Generalization challenges
[14]	2023	Mixed	IoT, Cloud Computing, AI, ML, Blockchain	Healthcare, Transportation, Smart Cities	Privacy, IoT vulnerabilities	Need for comprehensive frameworks
[15]	2023	Mixed	IoT	Smart Cities	Privacy	Generalization of findings
[16]	2021	Mixed	AI, IoT, and Cybersecurity tools	Cybersecurity and Industrial IoT	Vulnerability management, adversarial AI, and standardization	Scope, focus on literature, and generalization
[17]	2021	Mixed	IoT, IIoT	Healthcare, Smart Cities	Security, privacy	Biases in reviewed literature
[18]	2023	Quantitative	SNMP, KLD, ACL, MTD, and AWS	Healthcare, Transportation, Smart Cities	Security, privacy	Need for further validation,
[19]	2024	Quantitative	ML, Cloud Computing	Healthcare, Transportation, Smart Cities	Security measures	Generalization of findings
[20]	2022	Mixed	IoT, Cloud Computing, Blockchain	Healthcare, Urban Management	Privacy, security	Focus on existing research
[21]	2020	Qualitative	IoT, Cloud Computing, AI	Healthcare, Transportation, Smart Cities	Need for improved cybersecurity mechanisms	Complexity of IoT ecosystems
[22]	2024	Mixed	IoT, Cloud computing, ML, Blockchain	Healthcare, Transportation, Smart Cities	Privacy, security vulnerabilities	lack of comprehensive frameworks
[58]	2024	Mixed	Biometric Authentication, Blockchain	Healthcare, Smart Cities	Scalability, interoperability, resource constraints,	Focus on specific technologies, generalizability of findings
[61]	2023	Mixed	Biometric, OTP, Blockchain	IoHT	MFA applications in the IoHT	Lack of empirical data
[73]	2023	Qualitative	BBAC, PBAC	Healthcare, Smart Cities	Identity resolution	It does not cover all aspects of IoT security
[74]	2022	Quantitative	PUF, Edge Computing	Smart home	Privacy, scalability, interoperability	Complexity of implementation
[75]	2021	Quantitative	Blockchain, IoT, Edge Computing, Fog Computing	Smart Cities	Privacy, scalability	Limited scope of testing
[76]	2021	Mixed	Edge Computing, ML	Healthcare, Smart Cities, IoT	Data abstraction challenges	Need for further research into edge computing
[77]	2024	Quantitative	SDP, IAM, MSG	Cybersecurity, IoT	Challenges with implementing zero trust	Generalization of findings
[78]	2024	Mixed	Blockchain, FidAM, Biometric authentication, PUF	IoT	Scalability, resource constraints	Does not cover all authentication methods
[79]	2024	Quantitative	Lightweight Cryptography	IoT	Scalability	Scalability concerns
[80]	2023	Mixed	Hybrid Cryptography	Smart Cities, IoT	Security vulnerabilities	Performance evaluation
[81]	2023	Quantitative	MQTT, ChaCha20, AES	IoT	Lightweight cryptography	Focus on specific protocols
[82]	2024	Mixed	PKI	IoT	Scalability	Dependency on existing standards
[83]	2023	Mixed	Biometric	Smart Cities	Privacy, data security	Generalizability
[84]	2021	Mixed	Biometric	IoT	Challenges in biometric implementation	Lack of comprehensive coverage
[85]	2019	Quantitative	IoT	Healthcare, Transportation, Smart Cities	Need lightweight authentication mechanisms	Proposed mechanism may not apply to all IoT devices
[86]	2024	Mixed	Blockchain, Lightweight, Security Algorithms	IoT	Scalability	Dependency on PUFs
[88]	2024	Mixed	ML, Blockchain, IoT	Smart Cities	Scalability, interoperability, privacy	Lack of empirical data
[90]	2021	Mixed	Biometric, ML	Cybersecurity, IoT	Improving usability	Not cover all recent advancements
[91]	2020	Mixed	Blockchain, Fog Computing	IoT	Scalability, interoperability	Limited scope of experimentation
[95]	2023	Mixed	IoT	Smart Cities	Security, privacy	Scalability concerns
[89]	2019	Mixed	IoT	Smart Cities, transportation	Security challenges	Generalization of Findings

## 5. Future Directions

Future research should concentrate on creating more robust and adaptable authentication systems suited to the particular limitations of IoT environments. Based on the findings and limitations of the papers mentioned before and the analysis in Table 7, there is a need to develop lightweight authentication protocols designed for resource-constrained IoT devices and state that traditional authentication methods are impractical. Future research should focus on creating effective mechanisms that ensure robust security without compromising performance.

To improve real-time threat detection and response capabilities, research should investigate combining cutting-edge technology like AI and ML. Standardized protocols that guarantee interoperability across various IoT devices while upholding strong security must be developed. Future research could also examine how decentralized authentication techniques, like blockchain technology, might help with data integrity and trust concerns. Furthermore, to guide the development of user-friendly and efficient authentication frameworks that encourage user involvement and adherence, user-centered research is crucial to comprehending end users’ difficulties when putting safe authentication procedures into reality.

## 6. Conclusions

The authentication methods used for IoT devices in smart cities have been critically analyzed in this systematic literature review, emphasizing the need for enhanced security procedures given the explosive growth of linked technologies. The shortcomings of conventional authentication techniques become more noticeable as urban environments depend more and more on IoT technologies to boost productivity and service delivery. On the other hand, existing restrictions impact security and the wider uptake and effectiveness of IoT technologies.

Traditional authentication techniques do not scale well with the growing number of IoT devices, which can compromise the efficiency of smart city systems. Developing lightweight authentication protocols can improve scalability without sacrificing security. Another challenge is the inability of authentication methods to adapt to various environments, as IoT devices have different security requirements. Context-aware authentication systems that dynamically modify protocols according to environmental variables can improve both user experience and security. Interoperability issues also disturb IoT authentication as different manufacturers utilize incompatible protocols, leading to security vulnerabilities. Establishing universal authentication standards and collaborative frameworks can improve security and interoperability among various devices. Furthermore, cybersecurity risks, such as credential theft and MitM attacks, pose significant threats, and traditional approaches often lack resilience. Implementing MFA and leveraging ML-driven solutions can greatly improve protection and strengthen security across IoT devices. Future developments may involve blockchain-based decentralized authentication to lessen dependency on centralized servers while preserving device identity through distributed ledgers. While blockchain-based solutions, biometric systems, and lightweight protocols offer potential benefits, they also present significant drawbacks.

This paper establishes the foundation for future research aiming at creating creative and flexible authentication solutions designed to meet the unique needs of smart cities’ IoT by highlighting the gaps in existing knowledge and practice. The results highlight the value of a multifaceted strategy and support the use of cutting-edge technologies like edge computing and machine intelligence to improve security. Researchers, practitioners, and legislators must work together to develop robust authentication systems that safeguard user privacy and data integrity as smart cities develop. The review’s conclusions operate as a call to action for continued investigation and creativity to protect smart urban environments against new security risks in the future.

## Figures and Tables

**Figure 1 sensors-25-01649-f001:**
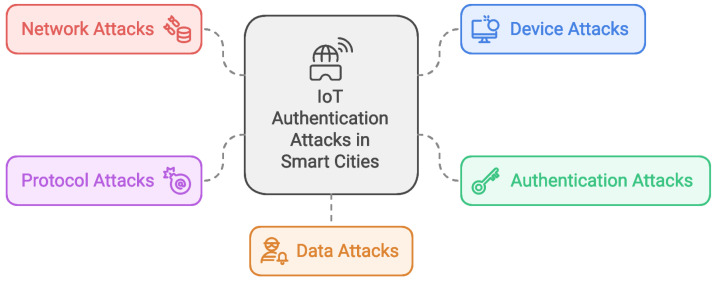
IoT authentication attacks in smart cities.

**Figure 2 sensors-25-01649-f002:**
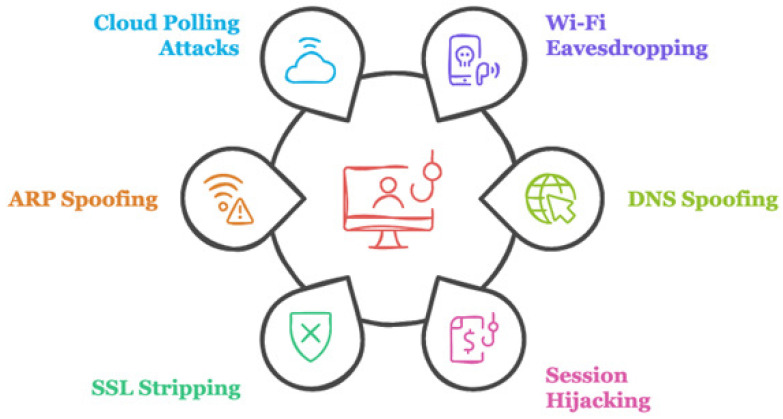
Types of MitM in IoT authentication.

**Figure 3 sensors-25-01649-f003:**
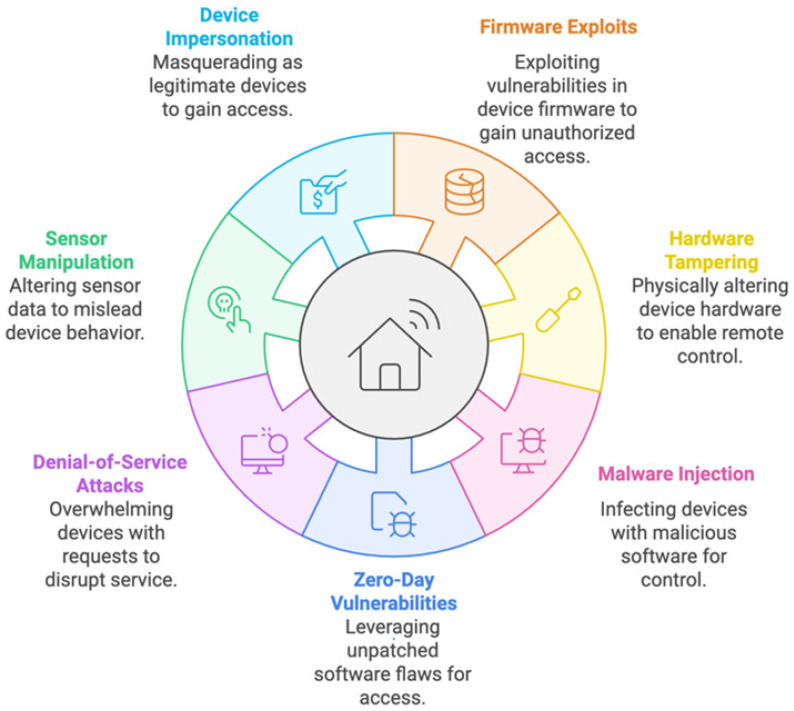
IoT device attack strategies.

**Figure 4 sensors-25-01649-f004:**
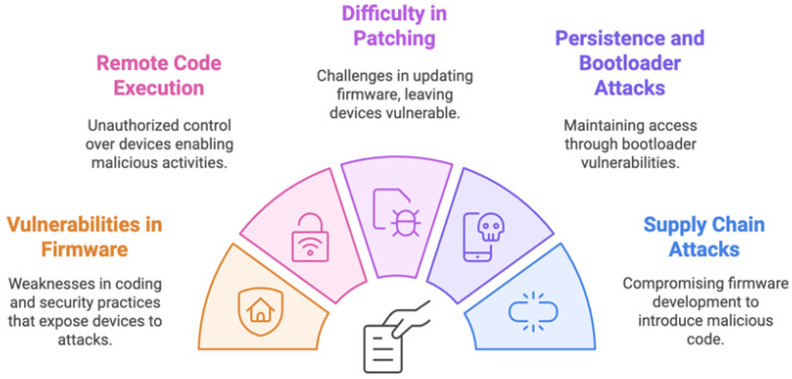
Firmware exploits.

**Figure 5 sensors-25-01649-f005:**
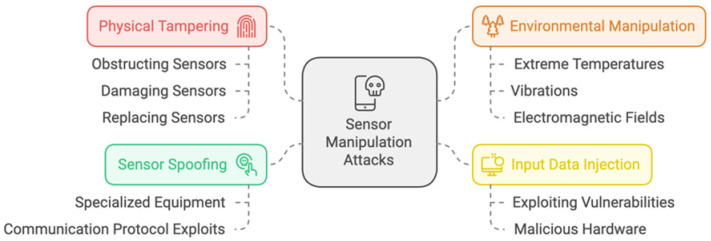
Techniques used in sensor manipulation attacks.

**Figure 6 sensors-25-01649-f006:**
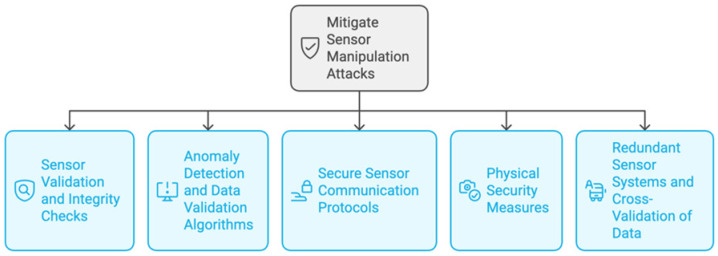
Mitigate sensor manipulation attacks.

**Figure 7 sensors-25-01649-f007:**
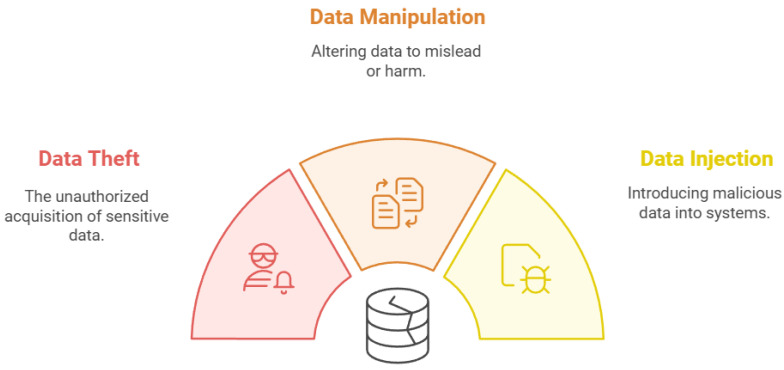
Types of data attacks.

**Figure 8 sensors-25-01649-f008:**
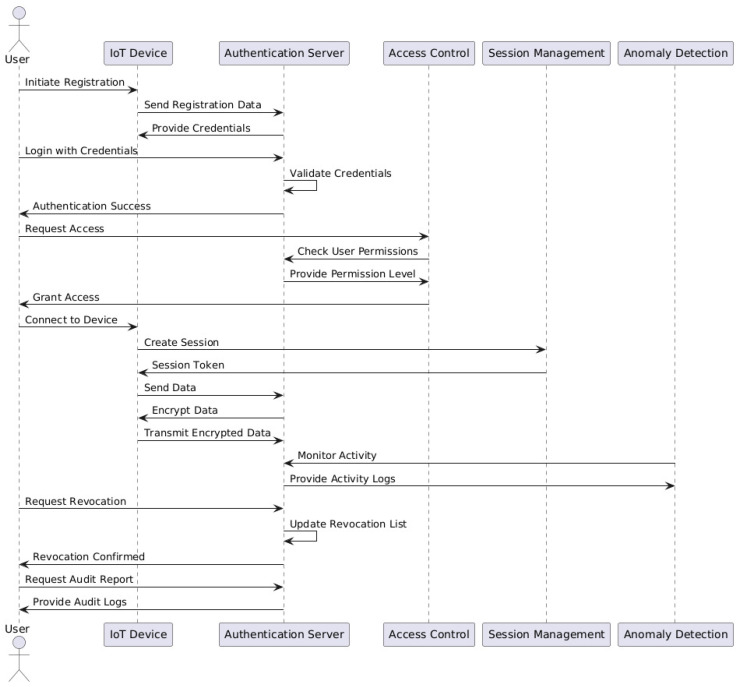
Authentication mechanism in smart cities.

**Figure 9 sensors-25-01649-f009:**
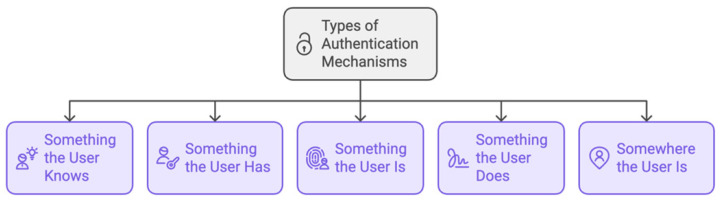
Types of authentication mechanisms.

**Figure 10 sensors-25-01649-f010:**
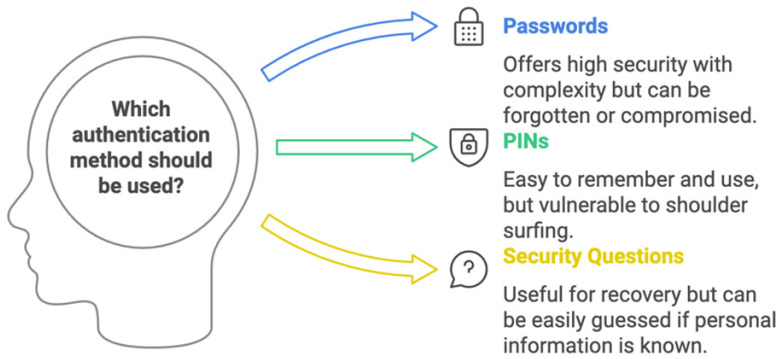
Examples of “something the user knows”.

**Figure 11 sensors-25-01649-f011:**
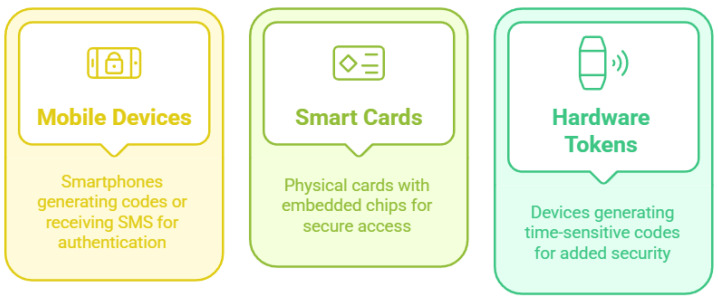
Examples of “something the user has”.

**Figure 12 sensors-25-01649-f012:**
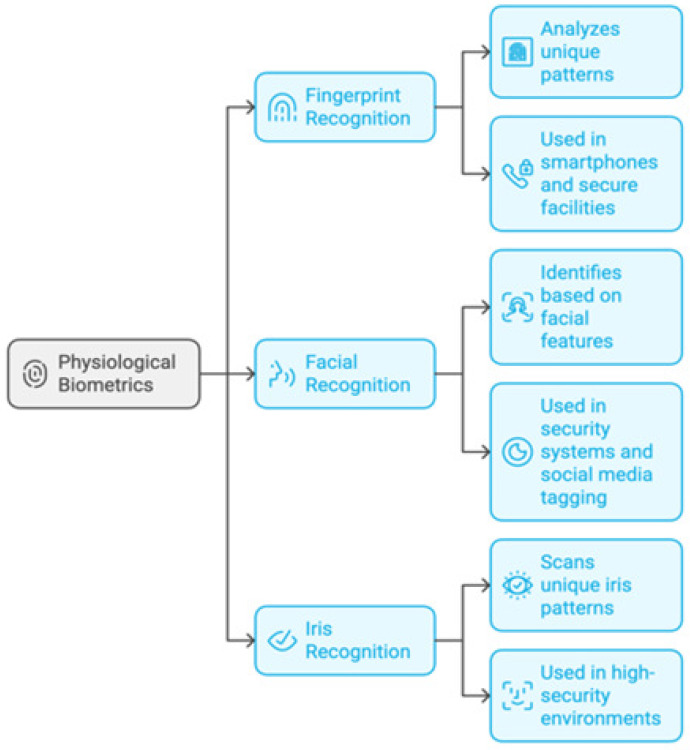
Examples of “something the user is”.

**Figure 13 sensors-25-01649-f013:**
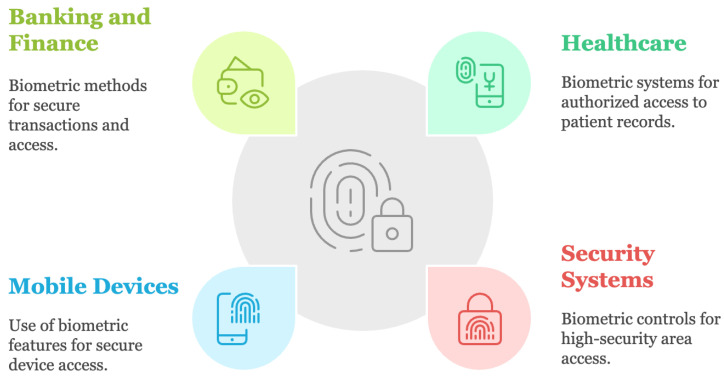
Applications of biometric authentication.

**Figure 14 sensors-25-01649-f014:**
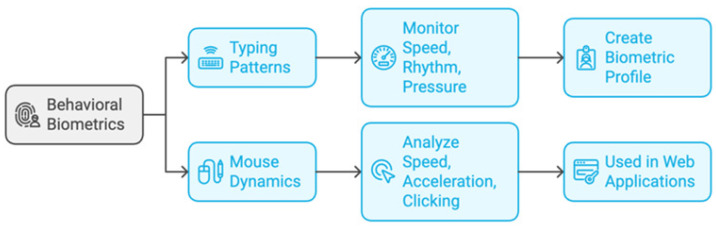
Examples of “something the user does”.

**Figure 15 sensors-25-01649-f015:**
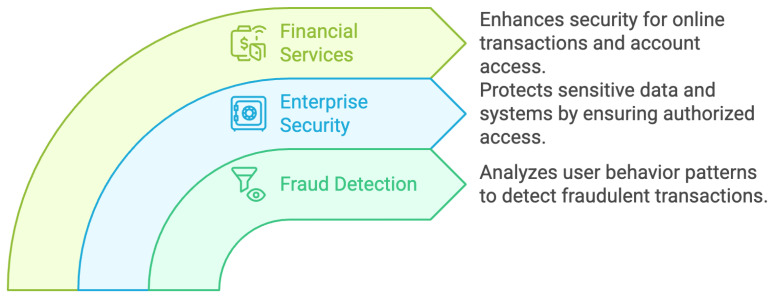
Applications of behavioral biometrics.

**Figure 16 sensors-25-01649-f016:**
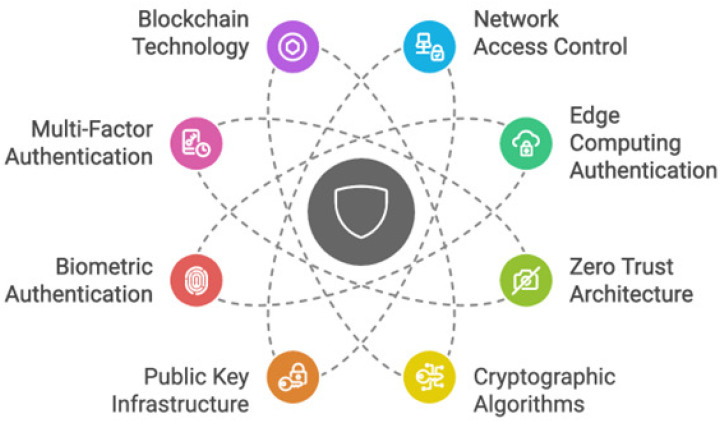
Tools and techniques for IOT authentication in smart cities.

**Figure 17 sensors-25-01649-f017:**
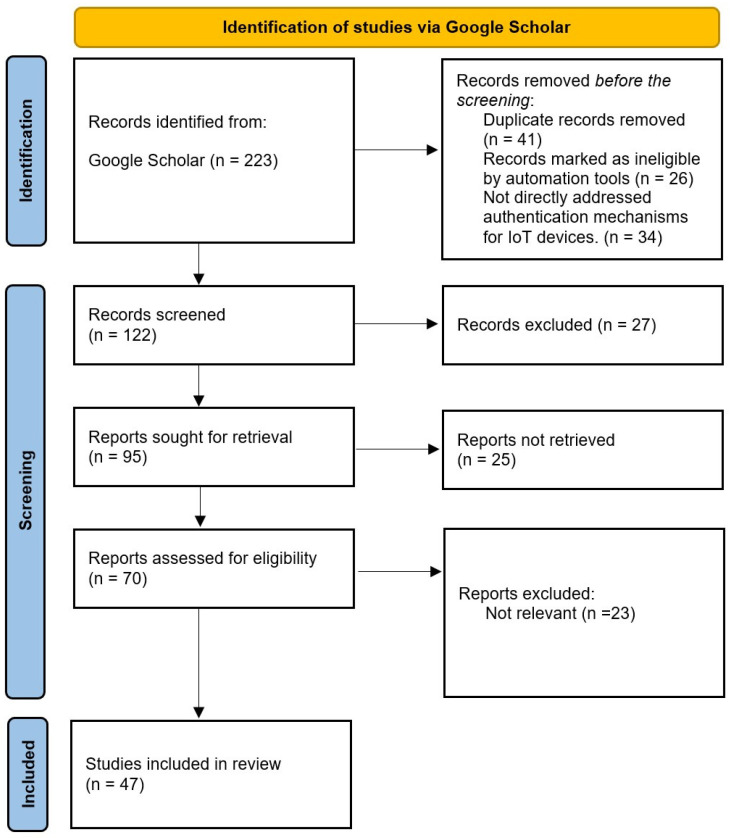
Selection of papers for literature review using PRISMA.

**Table 1 sensors-25-01649-t001:** Hardware tampering.

Hardware Tampering Type	Description
Physical Access	Gaining physical entry to devices.
Invasive Hardware Modifications	Direct manipulation of the components of the device.
Non-Invasive Attacks	Taking advantage of flaws in devices without making any physical alterations.
Firmware Modification	Changing the firmware of a device to gain control.
Supply Chain Attacks	Including harmful elements in the distribution process.
All Hardware Tampering Consequences	Possible consequences such as device takeover and data theft.

**Table 2 sensors-25-01649-t002:** Forms of DoS attacks.

DoS Attacks Forms	Description
Resource Exhaustion	Attackers may send a large number of requests that appear to be valid or corrupted data packets to the IoT device. When the device’s processing power is overloaded, it may become unresponsive or even crash, preventing authorized users from using the device.
Bandwidth Depletion	In this case, hackers may flood the IoT device with network traffic, using up all of the available bandwidth and blocking authorized connectivity. By doing this, the gadget may be successfully isolated from the other smart city equipment.
Authentication Exhaustion	To exhaust the resources of the IoT device, adversaries can try to authenticate with it repeatedly, taking advantage of flaws in the authentication procedure. For authorized users attempting to use the device, this may result in a denial of access.
Endpoint Saturation	The communication endpoints or gateways in charge of overseeing and verifying IoT devices may be the target of an attack. The devices may be unable to correctly authenticate and access the smart city systems if these crucial components are overloaded with requests.
Distributed DoS (DDoS)	In a more complex assault, enemies may use a botnet, a collection of hacked Internet of Things devices, to initiate a coordinated, extensive DoS attack. The impact on the targeted IoT infrastructure may be greatly increased as a result, possibly upsetting public safety systems, key services, or infrastructure monitoring and control.

**Table 3 sensors-25-01649-t003:** Types of authentication mechanisms.

Types	Examples	Advantages	Disadvantages
Something the User Knows	Passwords, PINs, security questions.	Simple, cost-effective, flexible.	Susceptible to social engineering, brute force, and phishing assaults. Users frequently select weak passwords or use the same ones on several websites.
Something the User Has	Smart cards, mobile devices, hardware tokens (e.g., YubiKey).	Offers an extra degree of protection and is more difficult to hack than passwords.	Needs users to carry an additional device, so it may be lost or stolen.
Something the User Is	Biometrics (fingerprints, facial recognition, iris scans).	Individually unique, hard to hack, and practical for consumers.	Needs certain technology and may result in false positives or negatives and privacy issues.
Something the User Does	Behavioral biometrics (typing patterns, mouse movements).	Include continuous authentication and less obtrusive operation due to its background operation.	Hard to execute well, and it may be impacted by shifts in user behavior brought on by several circumstances like sickness.
Somewhere the User Is	Geolocation-based authentication.	Can provide the authentication procedure with an extra layer of context.	Privacy issues, spoof ability, and unreliability due to fluctuating network circumstances.

**Table 4 sensors-25-01649-t004:** Most suitable mechanism.

Use-Case	Description
For High-Security Environments	The best option is frequently a mix of techniques by MFA. For example, a strong defense against unwanted access may be achieved by combining a mobile device for a one-time code with a password.
For User Convenience	A smooth user experience can be provided by biometric techniques, particularly on personal devices where security is less of an issue than on business systems.
For General Use	Security and user experience are well-balanced when the user has something they know and something they have, such as a password with SMS-based verification.

**Table 5 sensors-25-01649-t005:** Authentication technologies.

Technology	Scalability	Security	Latency
NAC	Moderate scalability because scaling across big networks might be challenging.	Increases security by regulating access according to regulations.	Although generally minimal latency, delays may be introduced during authentication checks.
Edge Computing	Extremely scalable as it minimizes the strain on central servers by enabling processing close to the data source.	Reduces data transfer hazards and localizes data processing to improve security.	Faster reaction times are made possible by low latency brought on by proximity to the data source.
ZTA	Extremely scalable and designed to accommodate expanding networks and gadgets.	Reduces trust assumptions and provides strong security by confirming each request.	This may cause some lag because verification procedures are ongoing.
Cryptographic Algorithms	Different algorithms have different levels of scalability, and as scale grows, some may need more resources.	Offers robust security, but it can be difficult to deploy and is susceptible to key compromise.	High latency is possible, particularly when dealing with intricate algorithms and huge data volumes.
PKI	Moderate scalability: a lot of certifications might make it complicated.	Robust security for authentication and data integrity, but administration is essential.	Because of certificate validation procedures, latency may be moderate.
Biometric Authentication	Generally scalable, and it may be used on a range of platforms and devices.	High security as it takes advantage of special biological characteristics. However, it is susceptible to spoofing.	Low latency: if the system is optimized, authentication is usually rapid.
MFA	Extremely scalable and connects with a variety of apps and systems with ease.	Because it uses several authentication techniques, it is extremely safe.	Latency ranges from low to considerable, depending on the techniques (e.g., SMS, applications).
Blockchain Technology	Scalability can be difficult; as the number of nodes grows, throughput may drop.	High security as a result of cryptographic methods and decentralized architecture.	High latency in general because of transaction verification and consensus processes.

**Table 7 sensors-25-01649-t007:** Analysis for related papers in authentication for IoT in smart cities.

Reference	Methodology	Finding	Authentication Techniques Used	Limitation
[3]	Conducted a thorough analysis of the current blockchain-based IoT device authentication systems.	Emphasized how important it is to find lightweight authentication methods because of the resource limitations of IoT devices.	Used blockchain technology to implement decentralized authentication mechanisms.	Acknowledged that the lack of resources in many IoT devices affects the suitability of some solutions.
[75]	Established API gateways that use smart contracts and blockchain technology to authenticate communications and confirm the authenticity of IoT devices.	Established that blockchain reduces the dangers associated with out-of-date firmware and improves the security and reliability of data from IoT devices in smart city applications.	Used Ethereum Blockchain Smart Contracts to validate and sign messages sent by IoT devices.	Affirmed that effective use of the solution depends on IoT devices being properly registered and validated, which is not always possible in real-world scenarios.
[96]	Suggested iTCALAS, an enhanced authentication technique that uses lightweight cryptographic operations, to overcome the shortcomings of the TCALAS scheme.	The suggested iTCALAS solution preserved scalability over numerous drone clusters while successfully preventing traceability and impersonation attacks.	Used lightweight symmetric key primitives in conjunction with temporal credentials to authenticate users and drones.	Asserted that the scheme’s efficacy depends on the cryptographic keys used in the authentication process being implemented and managed correctly.

## Data Availability

Not applicable.

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
