# Peer review of "A Review of the Authentication Techniques for Internet of Things Devices in Smart Cities: Opportunities, Challenges, and Future Directions"

_sensors, 2025, doi:10.3390/s25061649_

Round 1
Reviewer 1 Report
Comments and Suggestions for Authors
1. The length is too long, and there are many descriptive words with insignificant meaning in the multi-paragraph paragraphs.
2. Redundant expressions will show that the central theme of each paragraph is not obvious and prominent. Includes the abstract and the first, second and third parts.
Suggest condensed words and expressions more profound. Because the feeling of the full text is a summary of the existing situation, there is no refining and sublimation
Author Response
Dear Reviewer,
The attached file includes a point-by-point response.
Best regards,
Ahmed Aljughaiman

Reviewer 2 Report
Comments and Suggestions for Authors
The authors reviewed authentication techniques for IoT devices used in smart cities. They present a background, and issues of authentication in IoT systems. They proceed to discuss in some detail, attacks targeting authentication on IoT systems. They proceed to discuss the types of authentication mechanisms under five headings and also discuss tools and techniques used to achieve authentication. They presented related reviews, open challenges, and future directions before concluding the article. The article, however, lacks a comprehensive discussion of peer research in literature. Suggested revisions are listed below
1. The article lacks a review of peer research work in authentication. Under the various authentication mechanisms, there needs to be a discussion of peer research conducted and proposed solutions by peers in literature under the categories to present the current state of the art in authentication. This is a significant component of a review paper
2. A comparison of peer work (not only related reviews) is absent from the article. This is necessary for this article to provide the practical gaps and possible future directions in authentication for IoT systems/smart cities in literature
3. The third and fourth points under Research contributions mention the evaluation and assessment of authentication methods. The article, however, does not provide measurable information (eg. Performance information of strategies) to provide a practical assessment of authentication strategies. It would be helpful to include such information in the article.
4. Sections 3.1 and 3.2 are closely related. Consider merging the two or indicating under which authentication mechanisms in 3.1 the tools and techniques in 3.2 are used.
Author Response

(The authors gave the same response as above.)

Reviewer 3 Report
Comments and Suggestions for Authors
The work proposed for review has a contemporary theme. The authors have done a very in-depth and thorough research related to the security of IoT networks - in particular smart cities. The authors have reviewed 92 sources and they have been systematized in different areas - Attacks on IoT Authentication, Authentication Mechanism and Related Work. Finally, the authors conclude with an analysis to summarize the main issues related to the security of IoT networks.
Despite the fact that the proposed work does not offer anything new (there are many different works related to the literature review of the IoT network security problem) or any new scientific findings, it is interesting and valuable due to:
- the structure of the work and the quality of the presentation of the material (the authors present examples as well as possible solutions for each of the different security problems);
- the comprehensibility of the writing - no complicated expressions are used (it will be readable by readers who are not specialists in the subject).
Note to authors: some of the figures are vague (11 and 13) or very small and the text in them is not very easy to read (15, 17, 4, 5, 6, 8, 9). They should be edited to make them easier to read.
Best regards!
Author Response

(The authors gave the same response as above.)

Reviewer 4 Report
Comments and Suggestions for Authors
The authors present a systematic literature review on authentication techniques in IoT. The article is well structured, the English is good. The citations are adequate, although some more should be added. The first part of the article is a very good one. The review of attack techniques and authentication mechanisms in IoT is a great job. The second part can be easily improved.
Some improvements to the paper are below.
1. Introduction (section 1). There are no citations, some should be added.
2. Introduction (section 1.1). Smart lighting, smart parking systems in cities or smart traffic in public transport systems can be incorporated into the text.
3. Introduction (section 1.2). Add some citations in the first part, when MQTT and MiTM attacks are introduced.
4. Introduction (section 1.3). The second paragraph has no references.
5. Section 2 and 3. How were the papers chosen in these sections?
6. Line 293. Suggestion: better 'compromise' instead of 'jeopardize'
7. Figure 17. Why were the 32 papers rejected?
8. Related work. A new section should be included in which the papers are classified according to their content: types of attacks, types of authorization, for example. The section should start on line 791.
9. Section 5. The new parts could be removed and included in the conclusions. Other parts are already sufficiently explained in the article.
Author Response

(The authors gave the same response as above.)

Round 2
Reviewer 2 Report
Comments and Suggestions for Authors
The Authors have amply addressed the concerns raised.